# DAF-18/PTEN protects LIN-35/Rb from CLP-1/CAPN-mediated cleavage to promote starvation resistance

Jingxian Chen, Rojin Chitrakar, L Ryan Baugh

**Starvation resistance is a fundamental trait with profound influence on fitness and disease risk. DAF-18, the *Caenorhabditis elegans* ortholog of the tumor suppressor PTEN, promotes starvation resistance. PTEN is a dual phosphatase, and DAF-18 promotes starvation resistance as a lipid phosphatase by antagonizing insulin/IGF and PI3K signaling, activating the tumor suppressor DAF-16/FoxO. However, if or how DAF-18/PTEN protein-phosphatase activity promotes starvation resistance is unknown. Using genetic, genomic, bioinformatic, and biochemical approaches, we identified the *C. elegans* retinoblastoma/RB protein homolog, LIN-35/Rb, as a critical mediator of the effect of DAF-18/PTEN on starvation resistance. We show that DAF-18/PTEN protects LIN-35/Rb from cleavage by the *μ*-Calpain homolog CLP-1/CAPN, and that LIN-35/Rb together with the repressive DREAM complex promotes starvation resistance. We conclude that the tumor suppressors DAF-18/PTEN and LIN-35/Rb function in a linear pathway, with LIN-35/Rb and the rest of the DREAM complex functioning as a transcriptional effector of DAF-18/PTEN protein-phosphatase activity resulting in repression of germline gene expression. This work is significant for revealing a network of tumor suppressors that promote survival during cellular and developmental quiescence.**

## Introduction

One of the most fascinating things about biology is the ability of organisms to robustly adapt to different environmental conditions. Many animals enter developmental diapause or a diapause-like state to endure unfavorable conditions (Easwaran & Montell, 2023). In the nematode *Caenorhabditis elegans*, third stage larvae arrest development in the dauer diapause due to high population density, limited nutrient availability, and high temperature (Baugh & Hu, 2020). When *C. elegans* embryos hatch in the complete absence of food, they arrest development in the first larval stage (L1 arrest or L1 diapause) (Baugh, 2013). There is no cell proliferation, migration, or

fusion during L1 arrest (Baugh & Sternberg, 2006), and gene expression and metabolism are dramatically altered to support survival (Baugh et al, 2009; Hibshman et al, 2017; Webster et al, 2022). Larvae can survive L1 arrest for weeks (Johnson et al, 1984), they continue foraging, and they recover upon feeding, albeit with developmental delay (Olmedo et al, 2020) and reproductive costs (Jobson et al, 2015; Jordan et al, 2019) commensurate with the amount of time spent in arrest.

How long worms survive L1 arrest is regulated by a variety of conserved signaling pathways and processes, with insulin/insulin-like growth factor signaling (IIS) being critical (Fig 1A) (Baugh & Hu, 2020). During starvation, IIS is reduced, with activity of DAF-2/insulin-like growth factor receptor (IGFR) and downstream AGE-1/phosphoinositide 3-kinase (PI3K) signaling decreased. DAF-2/IGFR and AGE-1/PI3K antagonize the Forkhead box O (FoxO) transcription factor DAF-16 (Lin et al, 1997; Ogg et al, 1997), and DAF-16/FoxO nuclear localization and activity increase during L1 arrest (Weinkove et al, 2006; Hibshman et al, 2017). Nuclear DAF-16/FoxO activates transcription of genes that promote starvation resistance (Hibshman et al, 2017) and represses genes that promote postembryonic development (Kaplan et al, 2015). Consequently, loss-of-*daf-16* renders worms starvation sensitive, with compromised survival (Munoz & Riddle, 2003), and arrest-defective, with postembryonic development initiated in starved L1 larvae (Baugh & Sternberg, 2006). Mammalian FoxO proteins are tumor suppressors (Paik et al, 2007), and *daf-16*/*FoxO* suppresses tumors in *C. elegans* (Pinkston et al, 2006). Despite substantial impacts on phenotype, *daf-16*/*FoxO* does not account for much of the transcriptional response to starvation (Hibshman et al, 2017), suggesting IIS-independent regulation of transcription.

The tumor suppressor phosphatase and tensin (PTEN) is a potent negative regulator of IIS in humans and *C. elegans* (Chalhoub & Baker, 2009; Murphy & Hu, 2013). PTEN was originally called MMAC, because it is mutated in multiple advanced cancers at high frequency (Li et al, 1997; Steck et al, 1997). The sole *C. elegans* PTEN homolog DAF-18 is required for L1 arrest, and *daf-18*/*PTEN* mutants are arrest-defective (Fukuyama et al, 2006) and starvation sensitive (Fukuyama et al, 2012), like *daf-16* mutants, though more severe in both cases. *daf-18*/*PTEN* is also required to repress germline gene

---

Department of Biology, Duke University, Durham, NC, USA

Correspondence: ryan.baugh@duke.edu
Rojin Chitrakar's present address is Sequencing and Genomic Technologies Shared Resource, Durham, NC, USA

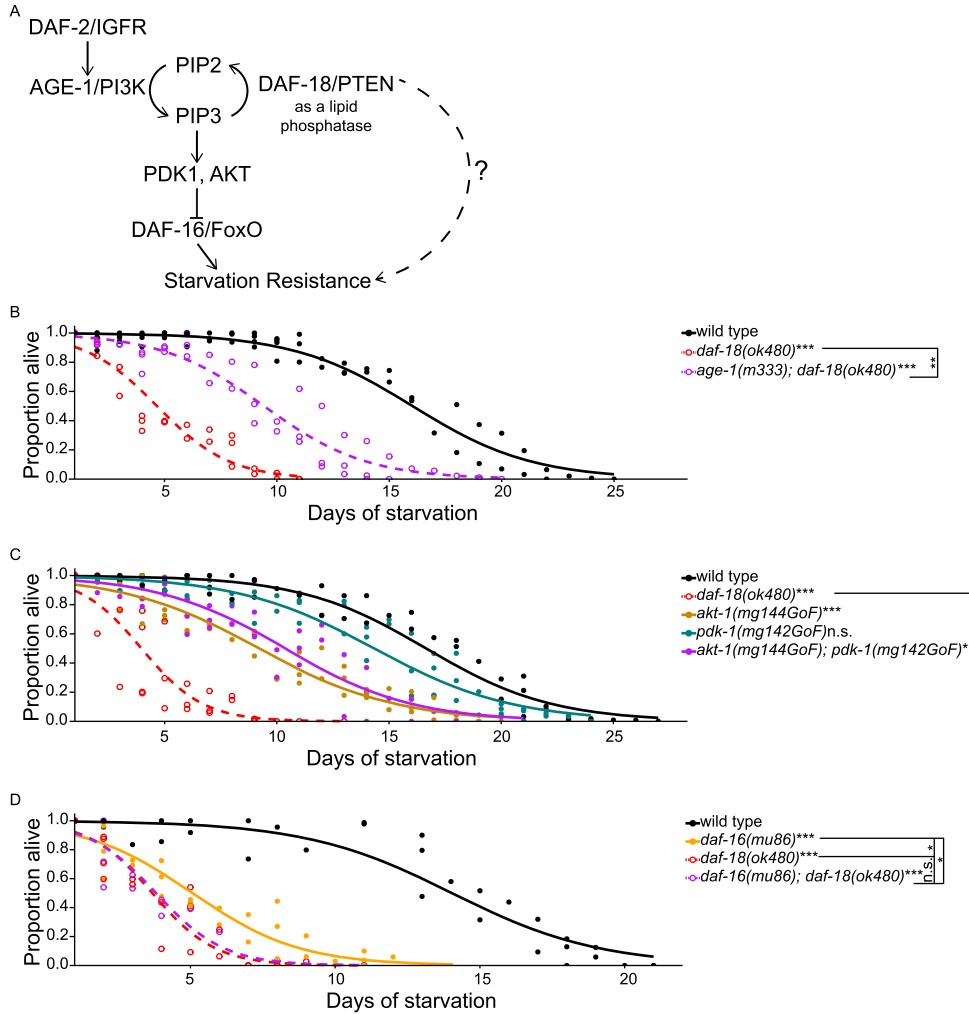

**Figure 1.  *daf-18/PTEN* promotes starvation resistance independently of AGE-1/PI3K signaling and *daf-16/FoxO*.** **(A)** Schematic summarizing relationships of genes analyzed in Fig 1. Dashed line and question mark denote that we sought to investigate *daf-18/PTEN*'s role independent of AGE-1/PI3K signaling. **(B, C, D)** Proportion of survivors is plotted throughout L1 starvation. Survival was scored daily. Four biological replicates were performed. Two-tailed, unpaired, variance-pooled *t* tests were performed on half-lives to compare genotypes. Unless otherwise noted with brackets, all comparisons were against WT. *$P < 0.05$; ***$P < 0.001$; n.s. not significant. For details on the statistical method, see the *Statistics for starvation survival* in the Materials and Methods section. **(B)** Number of scored animals was 105 ± 25 (mean ± SD). **(C)** Number of scored animals was 104 ± 31 (mean ± SD). **(D)** Number of scored animals was 105 ± 27 (mean ± SD). Half-lives of all four genotypes were subjected to Levene's test to assess variance homogeneity across groups, which suggested homogeneous variance. Half-lives were used in a two-way ANOVA (formula: half-life ~ loss of *daf-18* * loss of *daf-16*) to assess additivity of the effects of *daf-18* and *daf-16* on survival, which yielded an interaction *P*-value of $7 \times 10^{-6}$, indicating nonadditivity (interaction or dependence) of *daf-18* and *daf-16*.

expression during L1 arrest (Fry et al, 2021), but it is unclear how it exerts transcriptional repression. PTEN is best known for its lipid–phosphatase activity, which inhibits PI3K signaling by converting phosphatidylinositol-3, 4, 5-triphosphate (PIP3) to phosphatidylinositol-4, 5-bisphosphate (PIP2) in mammals and *C. elegans* (Myers et al, 1998; Ogg & Ruvkun, 1998). However, PTEN is also a protein-phosphatase (Myers & Tonks, 1997), and PTEN protein-phosphatase substrates with different roles in signaling and cell migration have been identified in mammals (Tamura et al, 1998; Gu et al, 1999, 2011; Shi et al, 2014; Abbas et al, 2019). It remains unclear whether DAF-18/PTEN functions as a protein phosphatase in *C. elegans*, if such activity contributes to regulation of L1 arrest, and, if so, how.

Like DAF-18/PTEN, LIN-35/Rb is a tumor-suppressor homolog that promotes survival in starved L1 larvae (Cui et al, 2013). *lin-35/Rb* also regulates vulva development along with other synthetic multivulva genes (Lu & Horvitz, 1998), and it represses expression of germline genes in the soma (Wang et al, 2005; Petrella et al, 2011; Wu et al, 2012; Rechtsteiner et al, 2019). LIN-35 is the sole *C. elegans* homolog of the human retinoblastoma (RB) pocket protein family (Lu & Horvitz, 1998). There are three RB-encoding genes in mammals, *RB1*, *RBL1*, and *RBL2*

(Henley & Dick, 2012). *RB1* was the first tumor suppressor to be cloned and characterized (Berry et al, 2019), and it was later found to be defective in many human cancers in addition to retinoblastoma (Burkhart & Sage, 2008). All three RB pocket proteins bind to adenovirus early region 2 binding factor (E2F) family transcription factors and the E2F dimerizing partner (DP) family members (Henley & Dick, 2012). Together with E2F, the protein encoded by *RB1*, pRB, enforces the G1/S cell cycle checkpoint through transcriptional repression (Uchida, 2016). In addition to E2F, the protein products of *RBL1* and *RBL2* (p107 and p130, respectively) recruit MuvB proteins (Guiley et al, 2015), and DP, RB, and E2F family proteins plus five MuvB proteins form the DREAM complex, which mediates transcriptional repression (Fischer et al, 2022). In *C. elegans*, mutation of *dpl-1/DP* or any of the genes encoding the five core MuvB/DREAM proteins results in ectopic expression of germline genes in the soma, like *lin-35/Rb* (Petrella et al, 2011; Wu et al, 2012). However, because LIN-35 is the sole pocket protein in *C. elegans*, it is unclear whether it regulates starvation resistance independently of the MuvB complex like pRB, as a component of the DREAM complex like p107 and p130, or via another mechanism.

DAF-18/PTEN and LIN-35/Rb are both important regulators of starvation resistance in *C. elegans*. However, whether there is a

functional link between them has not been addressed, nor has a connection between these two paramount tumor suppressors been reported in mammals. Here, we show that DAF-18/PTEN promotes starvation resistance independently of IIS, likely through its protein-phosphatase activity. Using a combination of genetics, functional genomics, and biochemistry, we discovered that *daf-18/PTEN* regulates LIN-35/Rb at the protein level to promote starvation resistance. LIN-35/Rb is cleaved and destabilized in the absence of *daf-18/PTEN*. We show that the human *µ*-Calpain protease homolog CLP-1/CAPN is responsible for this negative regulation of LIN-35/Rb and that *daf-18/PTEN* is a negative regulator of CLP-1/CAPN. We also report that the DREAM complex functions downstream of DAF-18/PTEN via LIN-35/Rb to promote starvation resistance likely by repressing germline gene expression. Our results provide a significant functional connection between DAF-18/PTEN and LIN-35/Rb that is likely conserved, and they identify a transcriptional effector mechanism of DAF-18/PTEN protein-phosphatase activity. Our insights into how these tumor suppressors promote survival during developmental quiescence have important implications for cancer, stem cell maintenance, and organismal fitness during starvation.

# Results

### *daf-18/PTEN* promotes starvation resistance independently of AGE-1/PI3K signaling and *daf-16/FoxO*

We used genetic analysis to examine whether DAF-18/PTEN functions independently of AGE-1/PI3K signaling to regulate L1 starvation resistance (Fig 1A). In contrast to *daf-18/PTEN* (Baugh & Sternberg, 2006; Fukuyama et al, 2012), mutation of *age-1/PI3K* increases L1 starvation resistance (Munoz & Riddle, 2003; Baugh & Sternberg, 2006; Fukuyama et al, 2012; Cui et al, 2013). There is no detectable PIP3 in *age-1* null mutants (Bharill et al, 2013), so we reasoned that if the only function of DAF-18/PTEN in this context is to counteract AGE-1/PI3K signaling by dephosphorylating PIP3 to produce PIP2 (as opposed to relying on protein-phosphatase activity), then loss of *age-1* should rescue resistance of an otherwise starvation-sensitive *daf-18* null mutant to WT levels or greater. Notably, the double null mutant *age-1(m333); daf-18(ok480)* (Morris et al, 1996; Brisbin et al, 2009) survived significantly longer than *daf-18(ok480)*, reflecting the role of the known DAF-18/PTEN lipid–phosphatase activity in promoting starvation resistance. However, this double mutant was significantly more starvation sensitive than WT (Fig 1B). The mothers of the double mutant were also homozygous double mutants (*daf-18[ok480]* suppresses the maternal Daf-c phenotype of *age-1[m333]*), and so the animals that were assayed were truly null for *daf-18* and *age-1*. This result suggests that DAF-18/PTEN promotes starvation resistance independently of AGE-1/PI3K signaling, potentially through its putative protein-phosphatase activity.

We assayed *akt-1/AKT* and *pdk-1/PDK* gain-of-function mutations to further examine AGE-1/PI3K-independent effects of DAF-18/PTEN. AGE-1/PI3K signaling activates PDK-1 and AKT-1 kinases, which antagonize DAF-16/FoxO effector function (Paradis & Ruvkun,

1998; Paradis et al, 1999). Both *pdk-1(mg142gf)* and *akt-1(mg144gf)* are strong gain-of-function alleles that completely suppress the *age-1* dauer-constitutive phenotype at 20°C (Paradis et al, 1999). We reasoned that if the only function of DAF-18/PTEN in promoting starvation resistance is to oppose AGE-1/PI3K signaling, then sufficiently increasing AGE-1/PI3K signaling activity should phenocopy *daf-18* null mutants. *akt-1(mg144gf)* significantly reduced starvation resistance, as expected, but not to the extent of the *daf-18* null allele *ok480* (Fig 1C). *pdk-1(mg142gf)* had no significant effect on its own, and the *pdk-1(mg142gf); akt-1(mg144gf)* double mutant was no more sensitive than *akt-1(mg144gf)* alone. Notably, the double mutant did not phenocopy *daf-18(ok480)*. These results could be due to inadequate activation of AGE-1/PI3K signaling with these gain-of-function alleles, but they are consistent with DAF-18/PTEN promoting starvation resistance independently of AGE-1/PI3K signaling.

We used genetic epistasis analysis to examine independence of *daf-16/FoxO* and *daf-18/PTEN* function. *daf-16/FoxO* is required for the *daf-2/IGFR* starvation-resistance phenotype (Baugh & Sternberg, 2006), suggesting it is the primary effector of AGE-1/PI3K signaling. However, a *daf-18* null mutant appears to be more sensitive to starvation than a *daf-16* null mutant, though they have not been analyzed together (Baugh & Sternberg, 2006; Fukuyama et al, 2012), suggesting that loss of DAF-18 activity does more than decrease DAF-16 activity via increased PI3K signaling. We confirmed that the null allele *daf-18(ok480)* is significantly more starvation sensitive than the null allele *daf-16(mu86)* (Fig 1D). Assuming *daf-16* is the only effector of AGE-1/PI3K signaling in this context, the greater starvation sensitivity of *daf-18(ok480)* than *daf-16(mu86)* suggests that loss of *daf-18* does more than release inhibition of AGE-1/PI3K signaling. Notably, the double mutant was not different from *daf-18(ok480)* alone, suggesting that loss of *daf-16* fails to decrease starvation sensitivity in a *daf-18* null background. Indeed, a two-way ANOVA revealed a significant statistical interaction between *daf-18* and *daf-16*, indicating nonadditivity of the two mutations. These results are consistent with DAF-18/PTEN inhibiting PI3K/AKT signaling via its lipid–phosphatase activity to activate DAF-16/FoxO, but they also support the conclusion that DAF-18/PTEN has an additional, independent function that promotes starvation resistance.

### Transcriptome-based epistasis analysis suggests *lin-35/Rb* mediates *daf-16/FoxO*-independent effects of *daf-18/PTEN*

We used mRNA sequencing (RNA-seq) to extend epistasis analysis of *daf-16/FoxO* and *daf-18/PTEN* to the transcriptome to isolate *daf-16*-independent effects of *daf-18* on gene expression. We performed bulk RNA-seq on *daf-16(mu86)* and *daf-18(ok480)* single null mutants, the *daf-16(mu86); daf-18(ok480)* double mutant, and WT in starved L1 larvae (Supplemental Data 1). We analyzed hatching in our staged populations and determined the timepoint when hatching first reached its maximum for all genotypes, and we collected our RNA-seq samples at that timepoint (Fig S1A; for details, see the *Hatching efficiency for determining RNA-seq sample collection timepoint* in the Materials and Methods section). This analysis suggested that each population was only ~4 h into L1 starvation on average, which is relatively early compared with the

peak of the starvation response at about 12 h (Baugh et al, 2009; Webster et al, 2022). Principal component analysis (PCA) of the RNA-seq data shows that all three mutants separate from WT in the first two principal components, which account for 44% of the total variance (Fig S1B), suggesting relatively robust effects on gene expression.

We used cluster analysis to get an overview of the RNA-seq results. We identified 871 genes that were differentially expressed across the four genotypes out of 15,018 detected genes, and we subjected the differentially expressed genes (DEGs) to hierarchical clustering (Fig 2A). Consistent with PCA (Fig S1B), the three mutants clustered and WT stood alone in the dendrogram (Fig 2A). Consistent with their starvation-resistance phenotypes (Fig 1D), the expression profiles of *daf-18(ok480)* and *daf-16(mu86); daf-18(ok480)* were more similar to each other than *daf-16(mu86)* (Fig 2A). *daf-16* and *daf-18* appear to have affected many genes in common, consistent with DAF-18/PTEN inhibiting AGE-1/PI3K signaling via its lipid–phosphatase activity. However, there also appear to have been many genes affected by *daf-18* but not *daf-16*, consistent with *daf-18* functioning independently of PI3K signaling and *daf-16*, potentially via its lipid–phosphatase activity.

We performed a transcriptome-wide epistasis analysis (Angeles-Albores et al, 2018) to formalize our interpretations of the RNA-seq data. In contrast to traditional epistasis, this analysis uses the genome-wide expression profile as the phenotype of interest. The rationale for this analysis is to use the results for each of the two single mutants compared with WT to generate expected results for the double mutant, assuming the mutants affect gene expression independently (log-additively). This was performed for the 563 genes that are significantly differentially expressed in all three mutant genotypes compared with WT, and bootstrapping and regression were used to determine a transcriptome-wide epistasis coefficient. We also simulated epistasis coefficients with our RNA-seq data using three predefined models for the relationship between *daf-18* and *daf-16*: linear/activation model where *daf-18* activates *daf-16*, and they are in a linear unbranched pathway; linear/suppression model where *daf-18* suppresses *daf-16*, and they are in a linear unbranched pathway; and a log-additive model where *daf-18* and *daf-16* act additively and independently of each other (in parallel) (predefined models are summarized in Fig 2B). We compared the observed distribution of epistasis coefficients generated with the parameter-free model to the three distributions resulting from the predefined models, computed model likelihoods with Bayesian statistics, and calculated odds ratios for the simulation results for each predefined model compared with the observed coefficients. Both the linear/suppression model and the log-additive model had an odds ratio of positive infinity, suggesting they are highly unlikely to represent the real relationship between *daf-18* and *daf-16* (Fig 2C). This is consistent with DAF-18/PTEN inhibiting AGE-1/PI3K signaling via its lipid–phosphatase activity to activate DAF-16/FoxO. However, the linear/activation model was rejected with an odds ratio of $2 \times 10^4$ (Fig 2C), which suggests that *daf-18/PTEN* affects transcriptional regulation by doing more than activating DAF-16/FoxO, consistent with cluster analysis (Fig 2A) and phenotypic analysis (Fig 1).

Differential gene expression analysis also suggested *daf-16/FoxO*-independent effects of *daf-18/PTEN*. We identified significantly DEGs

for each pair of genotypes (Supplemental Data 1). *daf-16/FoxO* and *daf-18/PTEN* shared a significant number DEGs compared with WT (Fig 2D), as expected, reflecting the lipid–phosphatase activity of DAF-18. However, *daf-18* had even more DEGs not in common with *daf-16* ("*daf-16*-independent targets of *daf-18*" highlighted in pink in Fig 2), suggesting a transcriptional effector in addition to DAF-16/FoxO.

The *daf-16*-independent targets of *daf-18* reflect the putative protein-phosphatase activity of DAF-18/PTEN. We subjected the *daf-16*-independent targets of *daf-18* to gene set enrichment analysis (GSEA) using WormExp, which compares a user-supplied gene set to a database of gene sets defined by published functional genomic experiments (e.g., RNA-seq with perturbation, ChIP-seq of specific proteins) (Yang et al, 2016). We found that genes from 12 experiments involving *lin-35/Rb* were significantly enriched (FDR < 0.1, $P = 6.7 \times 10^{-32}$ for the most significant enrichment; Supplemental Data 2). *lin-35/Rb* is known to promote L1 starvation resistance (Cui et al, 2013), but its activity has not been linked to *daf-18/PTEN*. We compared our *daf-16*-independent targets of *daf-18* to a set of "*lin-35* regulated targets" identified during L1 arrest (genes whose expression was affected by *lin-35* mutation in starved L1 larvae) (Cui et al, 2013), and confirmed significant overlap between these two gene sets (Fig 2E). These results reveal overlapping effects of *daf-18* (independent of PI3K signaling and *daf-16/FoxO*) and *lin-35/Rb* on gene expression during L1 arrest.

GSEA requires discrete gene sets, but we isolated *daf-16*-independent effects of *daf-18* for all genes by comparing *daf-16(mu86); daf-18(ok480)* and *daf-16(mu86)* and plotting cumulative distributions of $\log_2$ fold changes ($\log_2$FCs). "*lin-35*-activated targets" (genes down-regulated in *lin-35* mutant, starved L1 larvae compared with WT) (Cui et al, 2013) had significantly smaller $\log_2$FCs compared with all detected genes (Fig 2F). Conversely, "*lin-35*-repressed targets" (genes up-regulated in *lin-35* mutant, starved L1 larvae compared with WT) (Cui et al, 2013) had significantly larger $\log_2$FCs. We also compared our *daf-16*-independent targets of *daf-18* to "LIN-35 direct regulated targets" (activated or repressed; determined by RNA-seq and ChIP-seq in starved L1 larvae) (Gal et al, 2021), and found significant overlap (Fig 2G). "LIN-35 direct repressed targets" had significantly larger $\log_2$FCs (*daf-16; daf-18/daf-16*) compared with all detected genes, while "LIN-35 direct activated targets" were indistinguishable (Fig 2H). As a negative control, we performed the same GSEA between both *lin-35* target sets (Cui et al, 2013; Gal et al, 2021) and our "*daf-18*-independent targets of *daf-16*" (the 273 genes in Fig 2D), and neither was significantly enriched (Fig S1C and D). Our RNA-seq analysis suggests that *lin-35/Rb* and *daf-16/FoxO* independently affect gene expression during L1 arrest, and that *daf-18/PTEN* and *lin-35/Rb* function either in a linear pathway or in parallel with convergence on a common set of regulatory targets.

## *lin-35/Rb* functions downstream of *daf-18/PTEN* to promote starvation resistance

We used epistasis analysis to determine if *daf-16/FoxO* and *lin-35/Rb* function independently to promote L1 starvation resistance. *lin-35(n745)* and *daf-16(mu86)* null mutants were both significantly starvation sensitive compared with WT (Fig 3A). However, the double

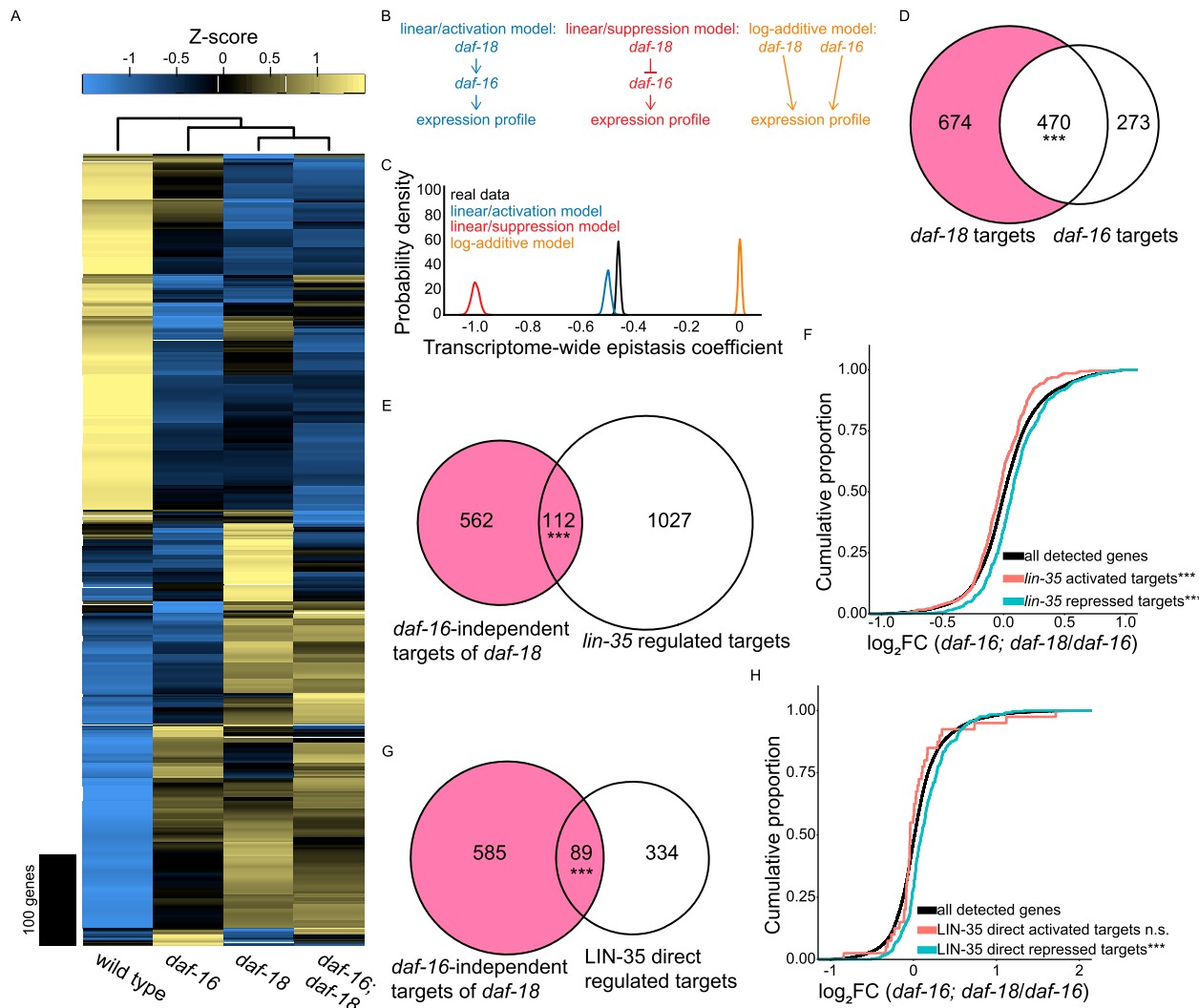

**Figure 2. Transcriptome-based epistasis analysis suggests *lin-35/Rb* mediates *daf-16/FoxO*-independent effects of *daf-18/PTEN*.**
**(A)** RNA-seq heatmap of genes whose expression was significantly different across WT, *daf-16(mu86)*, *daf-18(ok480)*, and *daf-16(mu86); daf-18(ok480)*. A generalized linear model found 871 genes out of 15,018 detected that were differentially expressed in the four genotypes tested. These 871 genes were z-score normalized and hierarchically clustered. Above the heatmap is the genotype dendrogram reflecting pairwise correlations between genotypes. For details on how these 871 genes were identified, see the *Differential expression analysis of RNA-seq data* in the Materials and Methods section. **(B)** Schematic summarizing predefined models for the transcriptome-wide epistasis analysis (Angeles-Albores et al, 2018). For more details, see the *Transcriptome-wide epistasis analysis* in the Materials in Methods section. **(C)** Distributions of transcriptome-wide epistasis coefficients are presented. RNA-seq data from WT, *daf-16(mu86)* and *daf-18(ok480)* single mutants, and *daf-16(mu86); daf-18(ok480)* double mutant were bootstrapped to simulate transcriptome-wide epistasis coefficients (defined in Angeles-Albores et al [2018]) under three predefined null models for the regulatory relationship between *daf-18* and *daf-16*. Data were also bootstrapped to determine transcriptome-wide epistasis coefficient under a parameter-free model for the real data. Odds ratios (ORs) were calculated by dividing the likelihood of the parameter-free model by each predefined null model. Model rejection: OR > $10^3$. Linear/activation model OR: $2 \times 10^4$; linear/suppression model OR: infinity; additive model OR: infinity. All three null models were rejected. For details on the statistics, see the *Transcriptome-wide epistasis analysis* in the Materials in Methods section. **(D)** Overlap of differentially expressed genes in *daf-18* versus WT and *daf-16* versus WT during L1 arrest. **(E)** Overlap between *daf-16*-independent targets of *daf-18* and *lin-35/Rb* targets in L1 arrest (from expression analysis in Cui et al [2013]). **(F)** *lin-35* target gene (Cui et al, 2013) expression changes in *daf-16; daf-18* versus *daf-16* are presented in cumulative distributions of log₂ fold-change (FC). *lin-35* activated and repressed targets (n = 488 and n = 651, respectively) are genes whose expression decreased or increased in the *lin-35* mutant compared with WT, respectively. **(G)** Overlap between *daf-16*-independent targets of *daf-18* and *lin-35* direct regulated targets in L1 arrest (determined by CHIP-seq and RNA-seq in Gal et al [2021]); i.e., genes bound by LIN-35 (direct) and whose expression was affected in the *lin-35* mutant (regulated). **(H)** *lin-35* direct regulated targets (Gal et al, 2021) gene expression changes in *daf-16; daf-18* versus *daf-16* are presented in cumulative distributions of log₂ fold-change (FC). N = 40 for LIN-35 direct activated targets. N = 383 for LIN-35 direct repressed targets. **(D, E, G)** The pink region represents *daf-16*-independent targets of *daf-18*. Hypergeometric tests were performed to assess the overlap significance between two gene sets, with the background being all detected genes in RNA-seq (see Supplemental Data 1). **(F, H)** The Kolmogorov–Smirnov tests were used to assess the equality between two cumulative distributions. All comparisons were against "all detected genes." **(D, E, F, G, H)** ***P < 0.001; n.s. not significant. See also Fig S1.

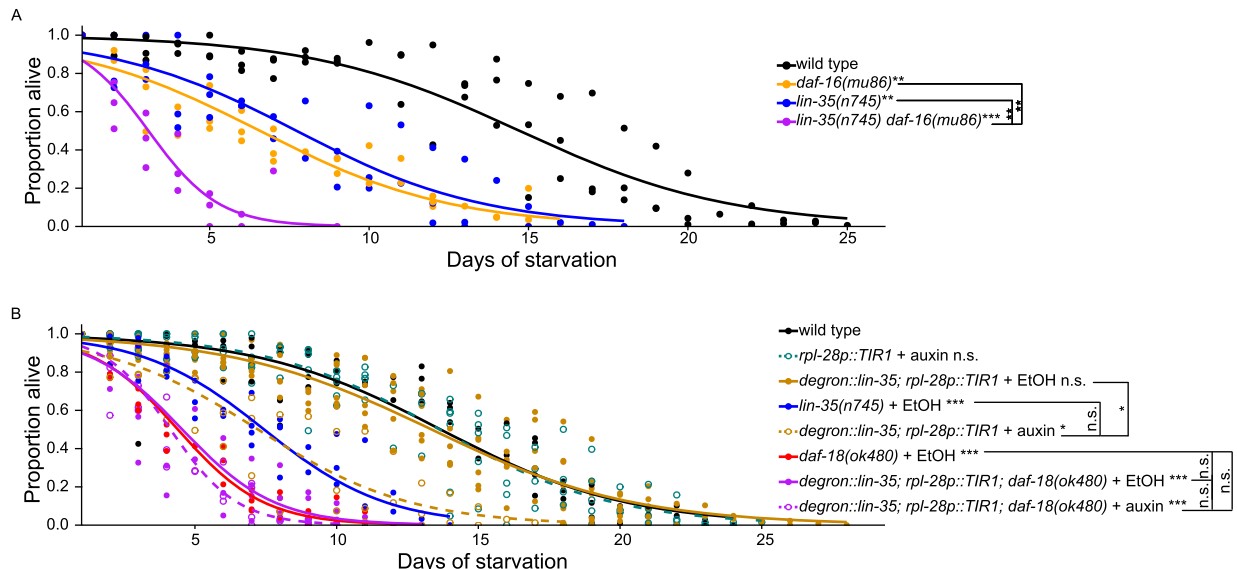

**Figure 3.** *lin-35/Rb* functions downstream of *daf-18/PTEN* to promote starvation resistance.
**(A, B)** Proportion of survivors is plotted throughout L1 starvation. Survival was scored daily. Unless otherwise noted with brackets, all comparisons were against WT. *$P < 0.05$; **$P < 0.01$; ***$P < 0.001$; n.s. not significant. For details on the statistical method, see the *Statistics for starvation survival* in the Materials and Methods section. **(A)** Three biological replicates were performed. Number of scored animals was 94 ± 28 (mean ± SD). Two-tailed, unpaired, variance-pooled *t* tests were performed on half-lives to compare genotypes. Half-lives of all four genotypes were subjected to Levene's test to assess variance homogeneity across groups, which suggested homogeneous variance. Half-lives were used in a two-way ANOVA (formula: half-life ~ loss of *lin-35* * loss of *daf-16*) to assess additivity of the effects of *lin-35* and *daf-16* on survival, which yielded an interaction *P*-value of 0.1, consistent with additivity (lack of interaction or independence) of *lin-35* and *daf-16*. **(B)** Seven biological replicates were performed. Number of scored animals was 96 ± 25 (mean ± SD). Two-tailed, unpaired, variance-unpooled *t* tests were performed on half-lives to compare conditions. Half-lives of *degron::lin-35; rpl-28::TIR1* +/− auxin and *degron::lin-35; rpl-28::TIR1; daf-18(ok480)* +/− auxin were subjected to Levene's test to assess variance homogeneity across groups, which suggested homogeneous variance. Half-lives were used in a two-way ANOVA (formula: half-life ~ loss of *daf-18* * auxin addition to degrade LIN-35) to assess additivity of the effects of *lin-35* and *daf-18* on survival, which yielded an interaction *P*-value of 0.003, suggesting nonadditivity (an interaction or dependence) of *lin-35* and *daf-18*. Auxin (indole-3-acetic acid) was used at 200 *μ*M and was prepared in ethanol (EtOH, the solvent). The *degron::lin-35* allele used is the same as the *degron::GFP::lin-35* allele used in Fig 4 (genotype abbreviated here). EtOH (solvent alone, no auxin) was used as control.

mutant was significantly more sensitive than either single mutant, as previously reported (Cui et al, 2013). A two-way ANOVA suggests no statistical interaction (additivity) between *lin-35* and *daf-16*, consistent with independent function downstream of *daf-18/PTEN*.

Our RNA-seq analysis revealed a positive correlation between the *daf-16*-independent effects of *daf-18/PTEN* and *lin-35/Rb* on gene expression, and we used epistasis analysis to determine whether they function in a common pathway. We found that *lin-35(n745); daf-18(ok480)* double null mutant is inviable without being starved, so we used the auxin-induced degradation system (Zhang et al, 2015) and *rpl-28*-promoter-driven TIR1 to degrade degron-tagged LIN-35 protein ubiquitously (Willis et al, 2021). Adding auxin (to early embryos before they hatch and enter L1 arrest) to *degron::lin-35; rpl-28p::TIR1* conferred starvation sensitivity indistinguishable from that of *lin-35(n745)*, suggesting that auxin-induced degradation resulted in potent degradation of LIN-35 protein and a null phenotype (Fig 3B). Notably, *daf-18(ok480)* was significantly more starvation sensitive than *lin-35(n745)*, consistent with DAF-18 activating DAF-16/FoxO via its lipid–phosphatase activity but also promoting starvation resistance through an AGE-1/PI3K and *daf-16*-independent mechanism involving *lin-35*. Critically, degrading degron::LIN-35 in a *daf-18* null mutant background did not have any effect. Furthermore, a two-way ANOVA analyzing the effects of degrading degron::LIN-35 and mutating *daf-18* suggests a

significant interaction (nonadditivity), as if *daf-18/PTEN* depends on *lin-35/Rb*. Taken together, Fig 3A and B support the conclusion that *lin-35/Rb* functions downstream of *daf-18/PTEN* to promote starvation resistance and that this effect is independent of AGE-1/PI3K signaling and *daf-16*.

### *daf-18/PTEN* protects LIN-35/Rb from cleavage

RNA-seq suggested that *daf-18* does not affect *lin-35/Rb* transcript abundance (Supplemental Data 1), so we asked if *daf-18/PTEN* could regulate LIN-35/Rb at the protein level. We collected whole-worm lysates of *degron::GFP::lin-35; rpl-28p::TIR1* (this strain was denoted as *degron::lin-35; rpl-28::TIR1* in Fig 3B for simplicity) in *daf-18(ok480)* and *daf-18(+)* background, and at the same timepoint as sample collection for RNA-seq (Fig S1A). We performed Western blots with a GFP antibody, and we observed the expected size of degron::GFP::LIN-35 at ~143 kD (Fig 4A). For the positive control, we added auxin to the starvation culture of *degron::GFP::lin-35; rpl-28p::TIR1* (the same way as we added auxin in Fig 3B), and we observed degradation of degron::GFP::LIN-35, as expected. The band for degron::GFP::LIN-35 also appeared dimmer in the *daf-18* mutant background. We blotted against alpha-tubulin as a sample-loading control, and we quantified alpha-tubulin-normalized degron::GFP::LIN-35 band intensity (Fig 4B). The degron::GFP::LIN-35 band intensity

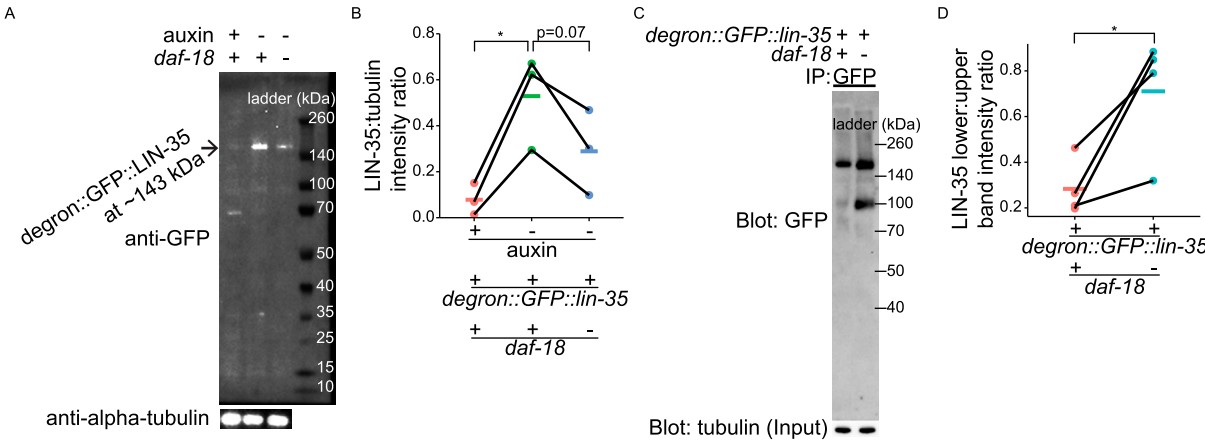

**Figure 4. *daf-18/PTEN* protects LIN-35/Rb from cleavage.**
**(A)** Representative Western blot showing the expression of LIN-35 in +/− auxin-mediated degradation and +/− *daf-18* conditions in starved L1 lysates (collected 16 h after hypochlorite treatment) is shown. Alpha-tubulin was used as a loading control. **(B)** Three biological replicates were performed (see (B)). **(B)** Quantification of LIN-35 expression level (normalized to alpha-tubulin) is plotted. Normalized LIN-35 expression levels were subjected to Bartlett's test to assess variance homogeneity across conditions. They were then used in two-tailed, paired, variance-pooled *t* tests to compare conditions, as Bartlett's test result suggested homogeneous variance. *P < 0.05. **(C)** Representative Western blot showing the expression of full-length and a shorter version of LIN-35 in +/− *daf-18* conditions following anti-GFP immunoprecipitation (IP) from starved L1 lysates (collected 16 h after hypochlorite treatment). IP inputs were normalized to have the same amount of total protein, as reflected by the alpha-tubulin blot. **(D)** Three biological replicates were performed (see (D)). **(D)** Quantification of the protein abundance ratio of shorter version LIN-35 and full-length LIN-35 in IP products. The ratios were subjected to Bartlett's test to assess variance homogeneity across conditions. They were then used in two-tailed, paired, variance-pooled *t* tests to compare +/− *daf-18*, as Bartlett's test result suggested homogeneous variance. *P < 0.05. **(A, B, C, D)** *daf-18 (−)* refers to *daf-18(ok480)*. The *degron::GFP::lin-35* allele used here is the same as the *degron::lin-35* allele used in Fig 3.
Source data are available for this figure.

significantly decreased compared with no auxin addition (Fig 4B). degron::GFP::LIN-35 abundance also decreased in *daf-18(ok480)* compared with the WT, but the *P*-value was 0.07 (Fig 4B; see Fig 5C). Nonetheless, these results suggest that DAF-18/PTEN promotes LIN-35/Rb stability during L1 arrest. Notably, loss of *daf-18* did not affect degron::GFP::LIN-35 abundance in fed L1 larvae (Fig S2A and B).

We used a GFP antibody to immunoprecipitate (IP) degron::GFP:: LIN-35 to investigate what happens to it in the absence of DAF-18/ PTEN. We blotted with a GFP antibody, and the expected full-length degron::GFP::LIN-35 band at ~143 kD was present (Fig 4C). However, to our surprise, a shorter band at ~100 kD was also evident in the *daf-18(ok480)* background. Quantification showed that enrichment of this smaller fragment in the mutant is statistically significant (Fig 4D). Together these results suggest that DAF-18/PTEN protects LIN-35/Rb from being cleaved during L1 arrest, which produces a shorter fragment and reduces abundance of full-length LIN-35/Rb.

### DAF-18/PTEN inhibits CLP-1/CAPN-mediated cleavage of LIN-35/Rb during starvation to support survival

*μ*-Calpain CAPN1 cleaves pRB after lysine 810 in human cervical cancer cell lines (Darnell et al, 2007; Tomita et al, 2020), suggesting that a Calpain homolog could cleave LIN-35/Rb in the absence of DAF-18/ PTEN. The closest *C. elegans* Calpain homolog to *μ*-Calpain is CLP-1/ CAPN (Fig S3A). We predicted a CLP-1-cleavage site after K541 in LIN-35 based on amino acid similarity to the *μ*-Calpain-cleavage site in human pRB (Fig 5A) (Tompa et al, 2004; Cuerrier et al, 2005; Darnell et al, 2007; Tomita et al, 2020). We subjected *degron::GFP::lin-35; rpl-28p::TIR1* with and without *daf-18(ok480)* IP lysates in Fig 4C and D to liquid

chromatography-tandem mass spectrometry (LC–MS/MS) and examined degron::GFP::LIN-35 peptide intensities in *daf-18(ok480)* versus WT. The results suggest an enrichment of N-terminal peptides (before K541) in *daf-18(ok480)* compared with WT (Fig S3B and C), which is consistent with Fig 4C and D, because GFP was added to the N-terminus of LIN-35.

We hypothesized that CLP-1/CAPN negatively regulates LIN-35/ Rb stability, so we asked if mutating *clp-1/CAPN* would rescue full-length degron::GFP::LIN-35 abundance in *daf-18(ok480)*. We collected whole-worm lysates of *degron::GFP::lin-35; rpl-28p::TIR1* with and without *daf-18(ok480)* and with and without *clp-1(tm858)* loss-of-function mutation, and we blotted against GFP. Surprisingly, and without explanation, full-length degron::GFP::LIN-35 abundance decreased in the *clp-1(tm858)* mutant compared with WT (Fig 5B and C). Full-length degron::GFP::LIN-35 abundance decreased in *daf-18(ok480)*, as expected (Fig 4A and B), and this time it was statistically significant (Fig 5B and C). Critically, this decrease in LIN-35 abundance in *daf-18(ok480)* was abolished in the *clp-1(tm858)* background (Fig 5B and C), supporting the conclusion that CLP-1/CAPN targets LIN-35/Rb for cleavage.

Our biochemical analysis suggests that *daf-18/PTEN* mutants are starvation sensitive in part due to decreased abundance of full-length LIN-35/Rb. We tested this hypothesis with a *lin-35* overexpression transgene (Fay et al, 2002), which modestly but significantly increased *daf-18(ok480)* starvation resistance but had no effect in a wild-type background (Fig S3D). We sought to further test this hypothesis by protecting LIN-35 from cleavage. Specifically, we hypothesized that blocking cleavage would not have an effect in a WT background but would rescue starvation sensitivity of *daf-18(ok480)*. Mutating *clp-1* nearly rescued *daf-18(ok480)* starvation sensitivity (*P* = 0.06), but it did

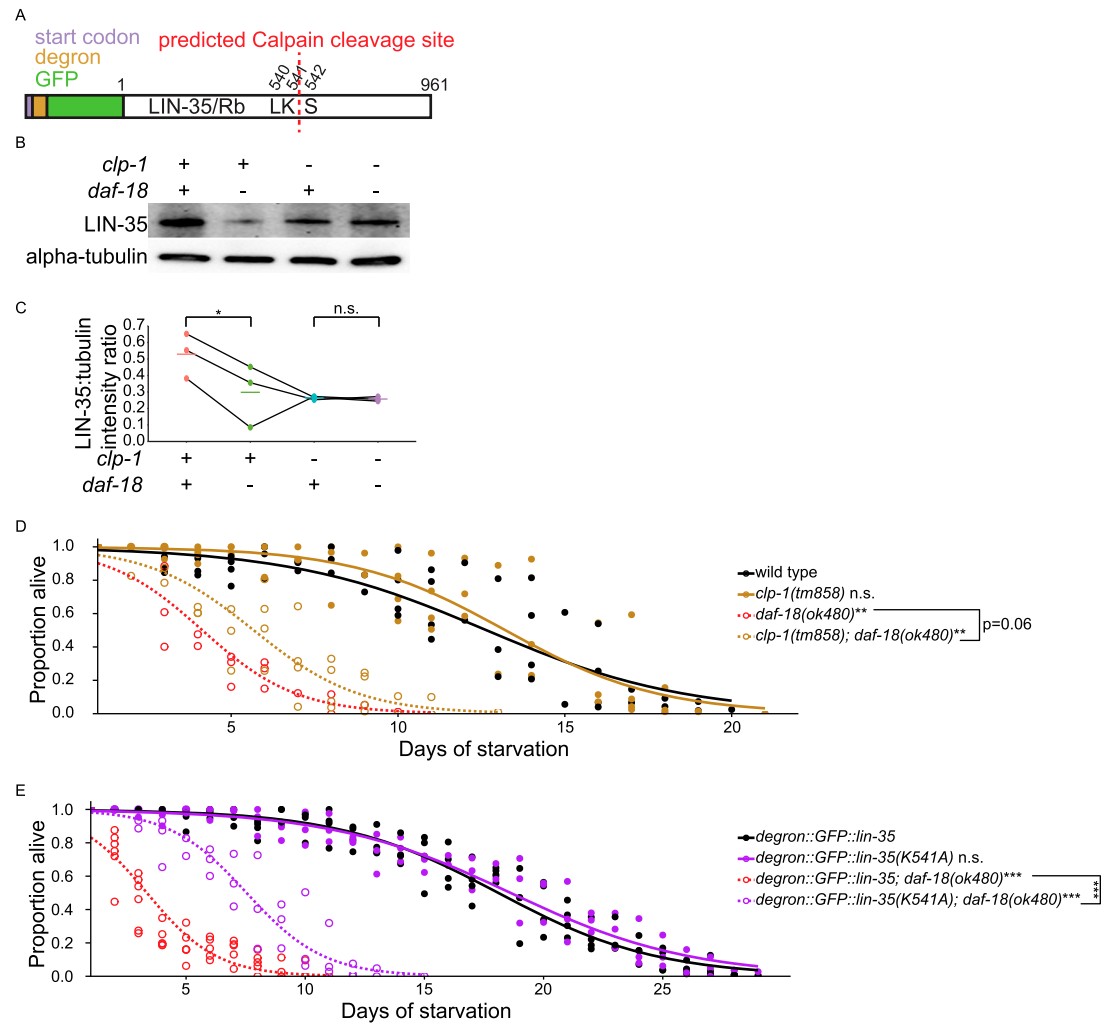

**Figure 5. DAF-18/PTEN inhibits CLP-1/CAPN-mediated cleavage of LIN-35/Rb during starvation to promote survival.**
**(A)** A schematic of degron:GFP::LIN-35 (strain from Willis et al [2021]) is presented. A Calpain-cleavage site was predicted based on amino acid similarity to the μ-Calpain-cleavage site in human pRB (Tompa et al, 2004; Cuerrier et al, 2005; Darnell et al, 2007; Tomita et al, 2020). Indicated amino acid positions are relative to native LIN-35/Rb. **(B)** A representative Western blot showing the expression of LIN-35 in +/− clp-1 and +/− daf-18 conditions in starved L1 lysates (collected 16 h after hypochlorite treatment) is shown. Alpha-tubulin was used as a loading control. **(C)** Three biological replicates were performed (see (C)). **(C)** Quantification of LIN-35 expression level (normalized to alpha-tubulin) is plotted. Normalized LIN-35 expression levels were subjected to Bartlett's test to assess variance homogeneity across conditions. They were then used in two-tailed, paired, variance-unpooled $t$ tests to compare conditions, as Bartlett's test result suggested heterogeneous variance. **(B, C)** daf-18 (−) refers to daf-18(ok480) and clp-1 (−) refers to clp-1(tm858). **(D, E)** Proportion of survivors throughout L1 starvation is plotted. Two-tailed, unpaired, variance-pooled $t$ tests were performed on half-lives to compare genotypes. Unless otherwise noted with brackets, all comparisons were against WT. For details on the statistical method, see the *Statistics for starvation survival* in the Materials and Methods section. **(D)** Four biological replicates were performed. Number of scored animals was 101 ± 30 (mean ± SD). **(E)** Six biological replicates were performed. Number of scored animals was 121 ± 26 (mean ± SD). **(C, D, E)** *$P$ < 0.05; **$P$ < 0.01; ***$P$ < 0.001; n.s. not significant. See also Fig S3. Source data are available for this figure.

not make a difference in WT (Fig 5D). We also created a point mutant that changes the predicted cleavage site lysine to alanine (K541A, in endogenous LIN-35 coordinates). A similar mutation in human *RB1* renders pRB resistant to Calpain-cleavage (Tomita et al, 2020). Notably, the cleavage-resistant mutant clearly and significantly rescued *daf-18(ok480)* starvation sensitivity, and it did not make a difference in WT (Fig 5E). These results demonstrate physiological significance of CLP-1/CAPN-mediated cleavage of LIN-35/Rb, supporting the conclusion that DAF-18/PTEN prevents CLP-1/CAPN from cleaving LIN-35/Rb during L1 arrest to promote survival.

## The DREAM complex represses transcription of germline genes downstream of *daf-18/PTEN* to support starvation survival

RB family proteins can repress transcription by forming a complex with E2F/DP transcription factors, and in some cases five MuvB proteins (LIN-52, LIN-9, LIN-54, LIN-37, and LIN-53) are further recruited to form the DREAM complex (Fischer & Muller, 2017; Fischer et al, 2022) (Fig 6A). LIN-35/Rb shares transcriptional targets and physically interacts with DREAM components (Harrison et al, 2006; Latorre et al, 2015; Goetsch et al, 2017; Gal et al, 2021; Goetsch &

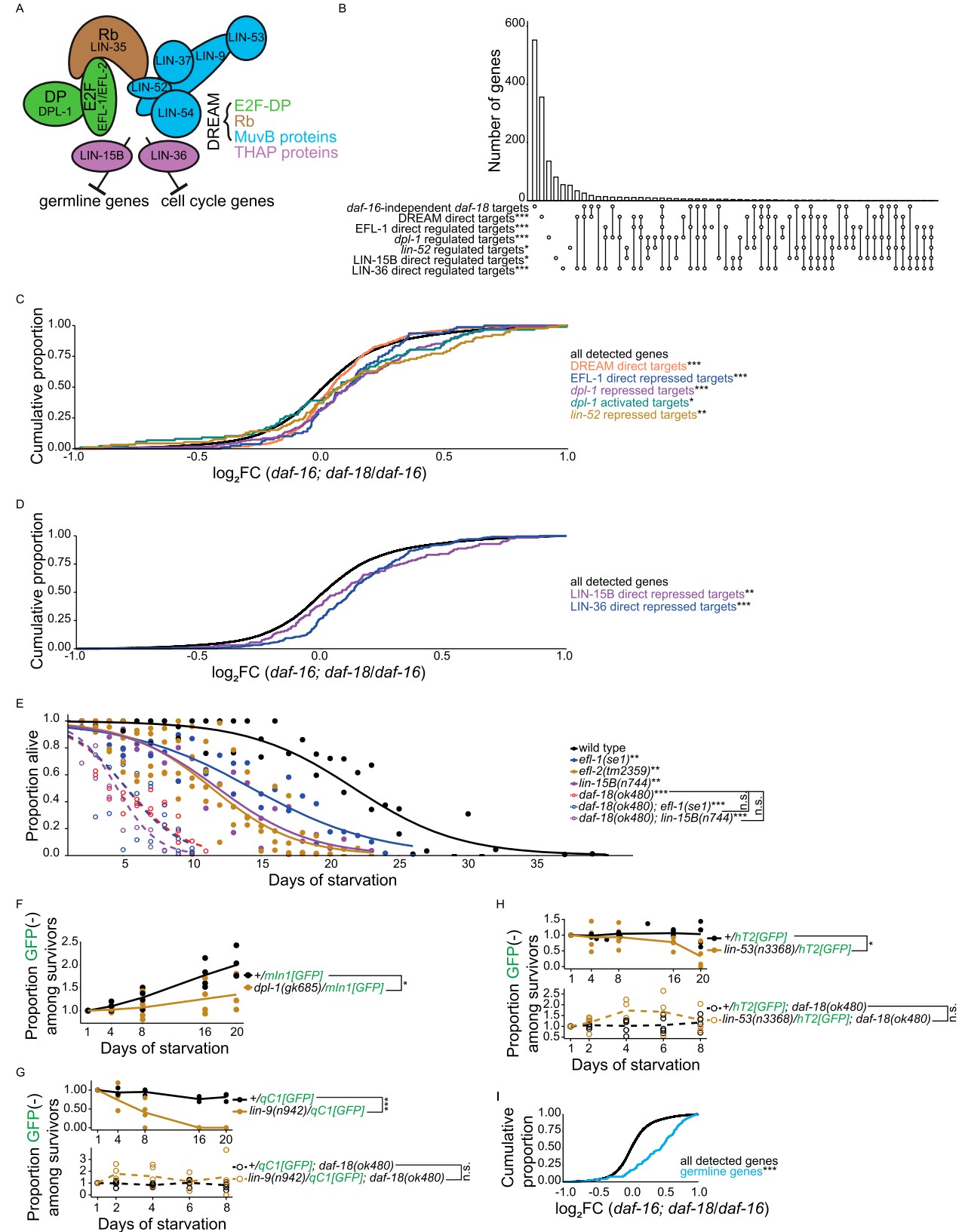

**Figure 6. The DREAM complex represses transcription of germline genes downstream of *daf-18*/PTEN to support starvation survival.**
**(A)** A schematic of the DREAM complex and THAP-domain proteins is presented (adapted from Gal et al [2021]; Goetsch & Strome [2022]). **(B)** Overlaps among *daf-16*-independent *daf-18* targets (defined in Fig 2D) and targets of DREAM, E2F, MuvB, and THAP-domain proteins are plotted. Single open circles indicate the number of genes

Strome, 2022). In addition, two THAP (Thanatos Associated Proteins) domain proteins are thought to mediate distinct functions of *C. elegans* DREAM, with LIN-36 mediating repression of cell cycle genes and LIN-15B mediating repression of germline genes in the soma (Gal et al, 2021) (Fig 6A). However, it is unclear if LIN-35/Rb functions with E2F/DP, DREAM, or either THAP-domain protein in promoting starvation resistance.

Given that *daf-18*/PTEN depends on *lin-35*/Rb to promote starvation resistance, we wanted to know whether the DREAM complex or just E2F/DP is a transcriptional effector of *daf-18*/PTEN. Together with RB family proteins, E2F/DP and DREAM repress transcription (Henley & Dick, 2012; Sadasivam & DeCaprio, 2013; Uchida, 2016), and DREAM is a transcriptional repressor in *C. elegans* (Petrella et al, 2011; Latorre et al, 2015; Goetsch et al, 2017; Rechtsteiner et al, 2019; Gal et al, 2021), so we hypothesized that loss-of *daf-18*/PTEN relieves E2F/DP or DREAM repression of *daf-18* targets independent of *daf-16*/FoxO. We revisited our RNA-seq data and found that *daf-16*-independent *daf-18* targets (defined in Fig 2D) are enriched for "EFL-1 direct regulated targets" (Gal et al, 2021) (Fig 6B) and that *daf-16; daf-18*/*daf-16* log$_2$FCs are greater for "EFL-1 direct repressed targets" than for all detected genes (Fig 6C). Furthermore, "*dpl-1* regulated targets" displayed similar enrichment (Fig 6B), and expression of "*dpl-1* repressed targets" was increased in the double mutant (Fig 6C). Together these results suggest that the *daf-16*-independent effects of *daf-18* depend on E2F/DP. "DREAM direct targets" (bound by eight *C. elegans* DREAM components) (Latorre et al, 2015) were enriched among *daf-16*-independent targets of *daf-18* (Fig 6B), and their expression was also significantly greater in the *daf-16; daf-18* double mutant than the *daf-16* single mutant (Fig 6C), suggesting that the *daf-16*-independent effects of *daf-18* also depend on DREAM. The *lin-52(3A)*

mutation was engineered to sever physical association between LIN-35/Rb and MuvB components of DREAM, and *lin-52(3A)*-up-regulated genes ("*lin-52* repressed targets" determined from RNA-seq) represent targets of repression by an intact DREAM complex (Goetsch & Strome, 2022). *lin-52* repressed targets are also enriched among *daf-16*-independent *daf-18* targets (Fig 6B), and their expression was significantly greater in the *daf-16; daf-18* double mutant than the *daf-16* single mutant (Fig 6C), further suggesting a role of DREAM in transcriptional repression downstream of DAF-18/PTEN. However, "*dpl-1* activated targets" also had higher expression in the *daf-16; daf-18* double mutant than the *daf-16* single mutant, albeit with a smaller effect size (Fig 6C). Increased expression of *dpl-1* activated targets is seemingly inconsistent with a unitary role of DREAM in repression, but the overlap could be because they are not all direct targets (they were determined by RNA-seq without CHIP-seq) (Gal et al, 2021), and so they may include secondary effects of alleviating transcriptional repression on direct targets. Taken together, these results support the hypothesis that DREAM functions as a transcriptional repressor downstream of DAF-18/PTEN.

We extended our analysis to the two THAP-domain proteins LIN-15B and LIN-36 (Gal et al, 2021). "LIN-15B direct regulated targets" and "LIN-36 direct regulated targets" are both enriched among *daf-16*-independent targets of *daf-18* (Fig 6B). In addition, both had larger *daf-16; daf-18*/*daf-16* log$_2$FCs than all detected genes (Fig 6D). Notably, a negative control gene set, "*daf-18*-independent targets of *daf-16*" (defined in Fig S1C and D), is not enriched for any of the six other gene sets included in Fig 6A (Fig S4A). These results suggest that in addition to intact DREAM, LIN-15B, and LIN-36 also contribute to transcriptional repression downstream of DAF-18/PTEN.

---

in each set that do not overlap with any other set, and vertical lines connecting open circles indicate the number of genes in the intersection of those gene sets. Targets activated and repressed by each factor are combined (based on RNA-seq analysis of mutants; "regulated"). "Direct" targets are based on ChIP-seq. Hypergeometric tests were performed to assess the overlap significance (indicated with asterisks) between *daf-16*-independent *daf-18* targets and each of the other gene sets, with the background being all detected genes in RNA-seq (see Supplemental Data 1). See (C, D) for published data sources. **(C)** Cumulative distributions of gene expression changes (log$_2$FCs) in *daf-16; daf-18* versus *daf-16* are plotted for DREAM direct target genes (n = 510; targets from Latorre et al [2015]), EFL-1 direct repressed targets (n = 79; determined by CHIP-seq and RNA-seq in Gal et al [2021]), *dpl-1* repressed and activated targets (n = 181 and n = 94, respectively; determined by RNA-seq in Gal et al [2021]), and *lin-52* repressed targets (n = 96; determined by RNA-seq in Goetsch & Strome [2022]). EFL-1 direct activated targets (n = 1) and *lin-52* activated targets (n = 13) were excluded because there were too few. **(D)** Cumulative distributions of gene expression changes in *daf-16; daf-18* versus *daf-16* are plotted for LIN-15B and LIN-36 direct repressed targets (n = 128 and n = 245, respectively; determined by CHIP-seq and RNA-seq in Gal et al [2021]). LIN-15B and LIN-36 direct activated targets (n = 15 and n = 5, respectively) were excluded because there were too few. **(C, D)** Kolmogorov-Smirnov tests were used to assess the equality of two cumulative distributions. All comparisons were against "all detected genes." **(E)** Proportion of survivors throughout L1 starvation is plotted. Survival was scored daily. Three biological replicates were performed. Number of animals scored was 112 ± 24 (mean ± SD). Two-tailed, unpaired, variance-pooled *t* tests were performed on half-lives to compare genotypes. Unless otherwise noted with brackets, all comparisons were against WT. Half-lives of WT, *efl-1(se1)*, and *lin-15B(n744)* single mutants, and corresponding *daf-18* double mutants, were subjected to Levene's test to assess variance homogeneity across groups, which suggested homogeneous variance. Half-lives were used in two-way ANOVA (formulae: half-life ~ loss of *daf-18* * loss of *efl-1*, half-life ~ loss of *daf-18* * loss of *lin-15B*) to assess additivity of the effects on survival of *daf-18* and *efl-1* and also *daf-18* and *lin-15B*, which yielded an interaction *P*-value of 0.002 for both tests, suggesting nonadditivity (an interaction or dependence) of *daf-18* with *efl-1* and *lin-15B*. **(F, G, H)** Proportion of GFP-negative worms (zygotic homozygous mutants) among survivors (normalized by Day 1) throughout L1 starvation is plotted. Parental genotypes are indicated in the legend. Survival was scored at five timepoints (days 1, 4, 8, 16, and 20 for strains without *daf-18[ok480]* and days 1, 2, 4, 6, and 8 for strains with *daf-18 [ok480]*). *daf-18(ok480)* mutants die rapidly during starvation (Fig 1), and strains carrying this allele could not be assayed beyond 8 d. **(F, G, H)** Proportion GFP(–) worms among survivors for each pair of genotypes in (F, G, H) (connected by brackets) was subjected to Levene's test to assess variance homogeneity across groups, which suggested heterogeneous variance. **(F, G, H)** Those proportions were subjected to a nonparametric two-way ANOVA (formula: proportion GFP[–] among survivors ~ genotype * duration of starvation) using the R package ARTool to assess whether *dpl-1(gk685)* in (F), *lin-9(n942)* in (G), and *lin-53(n3368)* in (H) displayed different levels of starvation resistance than WT. Asterisks represent *P*-values of the interaction between genotype and duration of starvation in nonparametric two-way ANOVA. **(F)** Four biological replicates were performed. Number of animals scored was 351 ± 123 (mean ± SD). Proportion GFP(–) worms among survivors for +/*mIn1[GFP]* was subjected to Bartlett's test to assess variance homogeneity across groups, which suggested heterogeneous variance. Those proportions were subjected to a nonparametric one-way ANOVA (formula: proportion GFP[–] among survivors ~ duration of starvation) using the kruskal.test function in R to assess whether proportion GFP[–] worms among survivors of +/*mIn1[GFP]* changed over time, which yielded a *P*-value of 0.003. **(G)** Number of animals scored was 399 ± 115 (mean ± SD). **(H)** Number of animals scored was 247 ± 78 (mean ± SD). **(G, H)** Five biological replicates were performed. **(I)** Cumulative distributions of log$_2$FCs in *daf-16; daf-18* versus *daf-16* are plotted for a high-confidence germline gene set (Fry et al, 2021) (denoted "germline genes"; n = 161). **(C, D, I)** The Kolmogorov-Smirnov tests were used to assess the equality between two cumulative distributions. All comparisons were against "all detected genes." **(B, C, D, E, F, G, H, I)** *P < 0.05; **P < 0.01; ***P < 0.001; n.s., not significant. See also Fig S4.

The preceding analysis revealed positive correlations between *daf-16*-independent effects of *daf-18* and a variety of DREAM components and mediators. We used epistasis analysis to determine whether E2F/DP, DREAM, and the THAP-domain proteins function downstream of *daf-18/PTEN*. Our model predicts that mutating DREAM components on their own causes a starvation-sensitive phenotype but that they do not affect the *daf-18* null mutant, like *lin-35/Rb* mutants. Loss-of-function mutants of *C. elegans* E2F genes, *efl-1(se1)* and *efl-2(tm2359)* were starvation sensitive (Fig 6E). Furthermore, *daf-18(ok480); efl-1(se1)* was no more sensitive than *daf-18(ok480)*, and the interaction between *daf-18* and *efl-1* was significant in a two-way ANOVA (Fig 6E), suggesting that *daf-18* depends on *efl-1/E2F* to promote starvation resistance. We analyzed the null mutant *dpl-1(gk685)*, which is inviable and must be maintained with a balancer chromosome. However, the GFP-marked balancer, *mIn1[GFP]*, caused starvation sensitivity by itself, as the proportion of WT worms (GFP[−] progeny of +/*mIn1[GFP]*) among survivors went up significantly over time (Fig 6F). Nevertheless, two-way ANOVA (interaction between genotype and duration of starvation) suggested that *dpl-1(gk685)* is starvation sensitive compared with WT (Fig 6F). These results reinforce the conclusion that *daf-18/PTEN* depends on E2F/DP to promote starvation resistance.

We analyzed two MuvB genes, *lin-9* and *lin-53*, to determine if *daf-18/PTEN* depends on MuvB/DREAM to promote starvation resistance. Like *dpl-1(gk685)*, the null mutants *lin-9(n942)*, and *lin-53(n3368)* are inviable and were maintained with a GFP-marked balancer, except the balancers used (*qC1[GFP]* and *hT2[GFP]*) did not cause starvation sensitivity on their own (Fig 6G and H). However, two-way ANOVA (interaction between genotype and duration of starvation) suggested that the proportion of homozygous *lin-9(n942)* and *lin-53(n3368)* worms among survivors went down over time (Fig 6G and H), suggesting that these MuvB mutants are starvation sensitive. Critically, *lin-9(n942)* and *lin-53(n3368)* were no more sensitive in a *daf-18(ok480)* null mutant background, as suggested by two-way ANOVA (interaction between genotype and duration of starvation) (Fig 6G and H). These results show that the positive correlations between *daf-18* and DREAM targets revealed in Fig 6B and C are functional, supporting the conclusion that DREAM functions downstream of DAF-18/PTEN to promote starvation resistance.

We analyzed *lin-36* and *lin-15B* to determine if *daf-18/PTEN* depends on either THAP-domain protein to promote starvation resistance. Interestingly, the *lin-15B(n744)* null mutant was starvation sensitive and nonadditive with *daf-18(ok480)*, but *lin-36(we36)* null mutants did not affect starvation resistance in the wild-type or *daf-18(ok480)* background (Fig S4B). These results suggest that LIN-35-DREAM promotes L1 starvation resistance downstream of DAF-18/PTEN via the THAP domain protein LIN-15B but that LIN-36 is dispensable, despite overlapping targets of LIN-36 and *daf-18* (Fig 6B and D).

Disruption of *daf-18/PTEN*, *lin-35/Rb*, *lin-37/MuvB*, and *lin-15B/THAP* results in upregulation of germline genes, and in the three latter cases this has been shown to be due to ectopic expression in the soma. We hypothesized that the *daf-16*-independent effects of *daf-18* (defined in Fig 2D), which are mediated at least in part through LIN-35/Rb and DREAM, subsume germline genes. We used a set of 161 "high-confidence germline genes" derived from the intersection of targets from four published germline gene sets (Fry et al, 2021) and found that they

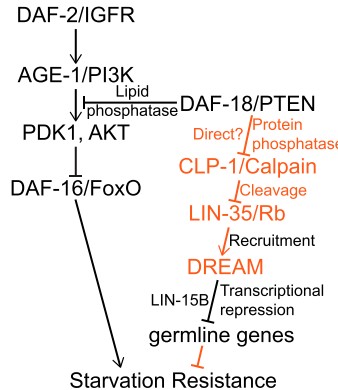

**Figure 7.   DAF-18/PTEN protein phosphatase acts through LIN-35/Rb and DREAM by inhibiting CLP-1/Calpain to promote starvation resistance.**
Proteins, regulatory relationships, and mechanisms known to regulate starvation resistance are in black, and those suggested by this study's results are in orange. See the Discussion section for details.

were significantly enriched among *daf-16*-independent targets of *daf-18* ($P = 8.4 \times 10^{-7}$). Notably, high-confidence germline genes were depleted among *daf-18*-independent targets of *daf-16* with near significance (depletion $P = 0.051$), suggesting a specific association between *daf-16*-independent targets of *daf-18* and germline genes. Moreover, *daf-16*; *daf-18/daf-16* $\log_2$FCs were significantly greater for high-confidence germline genes than for all detected genes (Fig 6I). These results suggest that DAF-18/PTEN acts through LIN-35/Rb and DREAM to repress germline gene expression during L1 arrest.

# Discussion

Starvation resistance is intimately related to human health and disease, but the molecular basis for it is not well understood. Here, we show that the tumor suppressor *daf-18/PTEN* promotes starvation resistance in *C. elegans* independent of its well-established regulation of PI3K and IIS, suggesting a critical function of DAF-18/PTEN protein-phosphatase activity (summarized in Fig 7). We discovered that DAF-18/PTEN protects another important tumor suppressor, LIN-35/Rb, from being cleaved by CLP-1/CAPN, permitting LIN-35/Rb to promote starvation resistance. LIN-35/Rb is the sole RB homolog/pocket protein in *C. elegans*, and we show that the DREAM complex and the THAP-domain protein LIN-15B are required for *daf-18/PTEN* to promote starvation resistance. Our results suggest that DREAM is a transcriptional effector of DAF-18/PTEN, and that DAF-18/PTEN promotes starvation resistance through its protein-phosphatase activity by repressing expression of germline genes via DREAM.

## PI3K and IIS-independent effects suggest DAF-18/PTEN protein-phosphatase activity promotes starvation resistance

PTEN is a potent tumor suppressor, and *C. elegans* *daf-18/PTEN* promotes developmental arrest and survival during L1 starvation

(Fukuyama et al, 2006, 2012). PTEN is both a lipid (Myers et al, 1998) and a protein phosphatase (Myers et al, 1997), but its protein-phosphatase activity has not been connected to starvation resistance or maintenance of cellular quiescence. DAF-18 is the sole PTEN homolog in *C. elegans*, and DAF-18 harbors a dual-phosphatase domain like PTEN (Chen et al, 2022), though protein-phosphatase activity has yet to be demonstrated. We show that loss of *daf-18/PTEN* reduces starvation resistance in an *age-1/PI3K* null background, that gain-of-function alleles effectively increasing PI3K signaling do not phenocopy *daf-18/PTEN* null alleles, and that loss of DAF-16/FoxO, the principal effector of PI3K/IIS in this context (Baugh & Hu, 2020), does not reduce starvation resistance in a *daf-18/PTEN* null background (Fig 1). These results suggest that DAF-18/PTEN protein-phosphatase activity promotes starvation resistance. However, we recognize that DAF-18/PTEN could possibly dephosphorylate a lipid other than PIP3 to account for these effects, and DAF-18/PTEN may have non-phosphatase regulatory activity. For example, PTEN interacts with histone H1 via its C-terminal tail to regulate chromatin condensation independent of its phosphatase activity (Chen et al, 2014), and PTEN interacts with but does not change the phosphorylation state of HSD17B8 (Zhao et al, 2024). Nonetheless, the simplest interpretation of our results together with prior knowledge is that DAF-18/PTEN lipid and protein-phosphatase activities both promote starvation resistance during L1 arrest.

It is desirable to use genetic analysis to dissect the function of *daf-18/PTEN* lipid and protein-phosphatase activities, and a pair of missense mutants have been presumed to specifically disrupt each of these two activities (Solari et al, 2005; Brisbin et al, 2009; Nakdimon et al, 2012; Zheng et al, 2018). However, biochemical analysis of homologous mammalian PTEN mutations revealed that these two missense mutations do not have such specific effects and instead disrupt both enzymatic activities (Furnari et al, 1998; Myers et al, 1998; Lee et al, 1999; Ramaswamy et al, 1999; Xiao et al, 2007; Rodriguez-Escudero et al, 2011). Likewise, these two missense mutations were engineered in *C. elegans* with genome editing, and complementation analysis revealed that they have overlapping effects on *daf-18/PTEN* function, affecting starvation resistance in L1 larvae and dauer formation (Chen et al, 2022; Wittes & Greenwald, 2022). Therefore, it is currently impossible to use genetic analysis to cleanly dissect the two phosphatase activities of PTEN or its homologs.

### DAF-18/PTEN protects LIN-35/Rb from CLP-1/CAPN-mediated cleavage during starvation

We identified LIN-35/Rb, another important tumor suppressor known to promote starvation resistance in *C. elegans* (Cui et al, 2013), as a mediator of PI3K/IIS-independent effects of DAF-18/PTEN (Figs 2 and 3). By using the expression profile from bulk RNA-seq analysis of null mutants affecting *daf-18/PTEN* and *daf-16/FoxO*, we were able to determine through multiple analyses that disruption of *daf-18/PTEN* affects gene expression independently of *daf-16/FoxO* and by extension PI3K and IIS (Fig 2). Moreover, by directly comparing a *daf-16; daf-18* double mutant and a *daf-18* mutant, we were able to isolate the putative effects of DAF-18/PTEN protein-phosphatase activity on gene expression during starvation. We used the resulting gene set ("*daf-16*-independent targets of *daf-*

18") to query a database of published results, discovering a significant positive correlation between the putative effects of DAF-18/PTEN protein-phosphatase activity and LIN-35/Rb on gene expression. We used epistasis analysis to confirm that *daf-16/FoxO* and *lin-35/Rb* function independently (Cui et al, 2013) and, critically, to demonstrate that *daf-18/PTEN* depends on *lin-35/Rb* to promote starvation resistance (Fig 3). These results suggest that *daf-18/PTEN* and *lin-35/Rb* promote L1 starvation resistance in a linear pathway. Notably, the *lin-35; daf-18* double null mutant was inviable, despite being well fed, suggesting independent function of these two pleiotropic genes in at least one other process impacting viability.

*lin-35/Rb* being epistatic to *daf-18/PTEN* suggested the possibility that LIN-35/Rb is directly dephosphorylated by DAF-18/PTEN, but multiple lines of evidence suggest that LIN-35/Rb is not a direct target of DAF-18/PTEN. We performed LC–MS/MS on starved L1 lysates after immunoprecipitating LIN-35/Rb to detect posttranslational modifications in WT and a *daf-18/PTEN* null mutant, and we did not detect an effect on LIN-35/Rb phosphorylation. We were also unable to detect LIN-35/Rb above background by LC–MS/MS or Western blot after immunoprecipitating DAF-18/PTEN from a *daf-18* "trapping mutant" background (Flint et al, 1997; Chen et al, 2022). These negative results prompted us to consider alternative hypotheses for how DAF-18/PTEN promotes LIN-35/Rb activity during L1 arrest.

We found that LIN-35/Rb is destabilized in the absence of DAF-18/PTEN in starved L1 larvae (Fig 4). Furthermore, a reduced molecular weight fragment of LIN-35/Rb was evident in the absence of DAF-18/PTEN, suggesting that DAF-18/PTEN protects LIN-35/Rb from cleavage. $\mu$-Calpain cleaves pRB in cervical cancer cells (Darnell et al, 2007; Tomita et al, 2020), so we investigated CLP-1/CAPN, the most similar *C. elegans* homolog of $\mu$-Calpain (Fig S3A). Multiple lines of evidence support the conclusion that DAF-18/PTEN antagonizes CLP-1/CAPN, which otherwise cleaves LIN-35/Rb to limit starvation resistance: LIN-35/Rb contains a predicted $\mu$-Calpain cleavage site at an appropriate location to account for the size of the LIN-35/Rb cleavage product, mutation of *clp-1/CAPN* rescues cleavage of LIN-35/Rb and also starvation resistance in the absence of DAF-18/PTEN, and, critically, a point mutation disrupting the predicted $\mu$-Calpain cleavage site in LIN-35/Rb rescues starvation resistance in a *daf-18/PTEN* null background (i.e., it is a gain-of-function allele) (Fig 5). These results are consistent with DAF-18/PTEN protein-phosphatase activity promoting starvation resistance via positive regulation of LIN-35/Rb, but they suggest that this regulation is indirect. However, it is unclear if in fact DAF-18/PTEN inhibits CLP-1/CAPN directly via dephosphorylation (Fig 7).

### DREAM functions as a transcriptional effector of DAF-18/PTEN to promote starvation resistance by repressing germline gene expression

*daf-18/PTEN* represses expression of germline genes during L1 starvation, but DAF-16/FoxO is not responsible, and the transcriptional effector of DAF-18/PTEN is unknown (Fry et al, 2021). It has also been unclear whether DAF-18/PTEN protein-phosphatase activity is involved in transcriptional regulation. Identification of the pocket protein LIN-35/Rb as a mediator of the effects of DAF-18/PTEN on starvation resistance suggested that either E2F/DP or the DREAM

complex could function as a transcriptional effector of DAF-18/PTEN during L1 arrest. We performed meta-analysis of various functional genomics datasets interrogating regulatory targets (determined with RNA-seq) and/or binding targets (determined with ChIP-seq) of E2F/DP and MuvB/DREAM components to determine whether the PI3K/IIS-independent effects of *daf-18/PTEN* may be mediated by E2F/DP alone or the entire DREAM complex (Fig 6A–C). Multiple components of E2F/DP and the rest of DREAM appeared to bind and regulate many of the same genes affected by the putative protein-phosphatase activity of DAF-18/PTEN, suggesting that DREAM is a transcriptional effector of DAF-18/PTEN. A similar analysis also suggested that the THAP-domain proteins LIN-36 and LIN-15B are involved (Fig 6D), consistent with them functioning as mediators of DREAM repression (Gal et al, 2021).

We used genetic analysis to determine if in fact E2F/DP, MuvB proteins (DREAM), and the THAP proteins function downstream of *daf-18/PTEN*. Several DREAM mutants have been assayed for their effects on L1 starvation resistance in *C. elegans* (Cui et al, 2013). *efl-1(se1)*, *dpl-1(n2994)*, and *lin-9(n112)* were assayed, but they are likely not null (Beitel et al, 2000; Ceol & Horvitz, 2001). Nonetheless, *efl-1(se1)* displayed starvation sensitivity, but *dpl-1(n2994)* and *lin-9(n112)* did not (Cui et al, 2013). This prompted us to examine a larger panel of DREAM mutants, including putative null alleles where available. We reproduced starvation sensitivity for *efl-1(se1)*, and putative null mutants *efl-2(tm2359)*, *dpl-1(gk685)*, *lin-9(n942)*, and *lin-53(n3368)* also conferred starvation sensitivity (Fig 6E–H). These results clearly suggest that the DREAM complex (LIN-35/Rb plus E2F/DP and MuvB proteins) promotes starvation resistance. Critically, the same mutations of *efl-1/E2F*, *dpl-1/DP*, *lin-9/MuvB*, and *lin-53/MuvB* did not affect starvation resistance in a *daf-18/PTEN* null background (Fig 6E–H), indicating that DAF-18/PTEN depends on the DREAM complex to promote starvation resistance. p107 has relatively high affinity for hsLIN-52/MuvB but pRB does not (Putta et al, 2022), and p107 and p130 are thought to function with DREAM while pRB is thought to function with E2F/DP but not MuvB (Henley & Dick, 2012). These observations together with our results suggest that LIN-35/Rb may share functional homology with the RB-like proteins p107 and p130 rather than pRB itself.

How does transcriptional regulation by DREAM, a repressor, promote starvation resistance? DREAM is well known for its role in regulating cell cycle genes (Fischer & Muller, 2017; Fischer et al, 2022), but DREAM also represses germline genes in the soma of *C. elegans* (Wang et al, 2005; Petrella et al, 2011; Wu et al, 2012; Rechtsteiner et al, 2019; Gal et al, 2021). Notably, LIN-35/Rb and DAF-18/PTEN also repress germline genes (Wang et al, 2005; Petrella et al, 2011; Wu et al, 2012; Rechtsteiner et al, 2019; Fry et al, 2021). Considering the two THAP-domain proteins, LIN-15B is thought to mediate DREAM repression of germline genes in the soma, and LIN-36 is thought to mediate repression of cell cycle genes (Gal et al, 2021). We assayed *lin-15B* and *lin-36* null mutants (Lu, 1999; Gal et al, 2021), and we found that *lin-15B* promotes starvation resistance (Fig 6E) but that *lin-36* does not (Fig S4B). Furthermore, loss of *lin-15B* does not affect starvation resistance in a *daf-18/PTEN* null mutant background (Fig 6E), indicating that DAF-18/PTEN depends on LIN-15B, but not LIN-36, in addition to DREAM to promote starvation resistance. This result and the fact that DAF-18/PTEN, LIN-35/Rb, DREAM, and LIN-15B all repress germline gene expression suggest that such repression supports starvation resistance. We therefore analyzed a set of "high-confidence germline genes" (Fry et al, 2021), and we found that DAF-18/PTEN

represses germline gene expression through its putative protein-phosphatase activity ("*daf-16*-independent targets of *daf-18*") but not it's lipid–phosphatase activity ("*daf-18*-independent targets of *daf-16*) (Figs 6I and S4C). Taken together, these results support a model in which the protein-phosphatase activity of DAF-18/PTEN protects LIN-35/Rb from CLP-1/CAPN-mediated cleavage, and that intact LIN-35/Rb recruits DREAM to germline genes for repression during starvation (Fig 7). *daf-18/PTEN* represses germline gene expression in the germ line, but the high-confidence germline genes used for analysis are not expressed exclusively in the germline (Fry et al, 2021). Given ectopic expression of germline genes in the soma of mutants affecting *lin-35/Rb, MuvB* components of DREAM, and *lin-15B/THAP* (Wang et al, 2005; Petrella et al, 2011; Wu et al, 2012; Rechtsteiner et al, 2019), we propose that DREAM represses germline gene expression in the germline and soma downstream of DAF-18/PTEN during L1 arrest (Fig 7).

Ectopic expression of germline genes following disruption of DREAM complex mutants commences during embryogenesis and continues in fed larvae (Petrella et al, 2011), suggesting the novel pathway we have defined downstream of DAF-18/PTEN could function outside of L1 arrest. DAF-18/PTEN may also contribute to transcriptional regulation of germline genes in embryos and fed larvae. In addition to repressing expression of germline genes during L1 arrest, *daf-18* enforces cell cycle arrest of the primordial germ cells (PGCs) during L1 arrest (Fry et al, 2021). *daf-18* also represses PGC divisions in fed larvae, as evidenced by precocious divisions in *daf-18* mutants after hatching in the presence of food. Furthermore, marks of transcriptional activation are evident before PGC divisions in *daf-18* mutant larvae and late embryos, and inhibiting transcription inhibits PGC divisions in this mutant (Fry et al, 2021). However, we found that LIN-35/Rb is not destabilized by loss of *daf-18* in fed L1 larvae as it is in starved L1 larvae (Figs 4 and S2). Although DREAM represses germline gene expression in fed larvae, this result suggests that DAF-18/PTEN regulation of LIN-35/Rb and DREAM is restricted to starved larvae, leaving open the question of how LIN-35/Rb and DREAM are regulated in fed larvae.

This study is significant as it reports a novel regulatory relationship between two important tumor suppressors, DAF-18/PTEN and LIN-35/Rb. Furthermore, it illustrates the critical function of these tumor suppressors along with DAF-16/FoxO beyond repressing proliferation in promoting survival of quiescence, both developmental and cellular. These insights improve understanding of the regulatory network governing starvation resistance and inform development of interventions to mitigate cancer, diabetes, and aging.

## Materials and Methods

### Strains used in this study

WT N2 is from the Sternberg Lab collection. AWR58 *lin-35(kea7[lin-35p::degron::GFP::lin-35]) I; keaSi10(rpl-28p::TIR1::mRuby::unc-54 3′UTR + Cbr-unc-119[+]) II* is from the Reinke Lab at University of Toronto. IC166 *daf-18(ok480) IV* is from the Chin-Sang Lab at Queen's University. JA1850 *lin-36(we36) III* is from the Ahringer Lab at University of Cambridge. LRB447 *lin-35 (n745) I* is from the Fay Lab at University of Wyoming and was a five-time backcrossed version of MT10430 *lin-35 (n745) I*. BQ1 *akt-*

1(mg306) V, CF1038 daf-16(mu86) I, GR1310 akt-1(mg144gf) V, GR1318 pdk-1(mg142gf) X, JJ1549 efl-1(se1) V, MT15107 lin-53(n3368) I/hT2 [bli-4(e937) let-?(q782) qIs48] (I;III), MT2495 lin-15B(n744) X and VC1523 dpl-1(gk685)/mIn1 [mIs14 dpy-10(e128)] II are from the Caenorhabditis Genetics Center (CGC). FX00858 clp-1(tm858) III and FX02359 efl-2(tm2359) II are from the National BioResource Project (NBRP):: C. elegans in Japan.

## Strains generated in this study

PHX8778 lin-35(kea7[lin-35p::degron::GFP::lin-35] syb8778[K541A]) I was generated by SunyBiotech.

LRB378 daf-16(mu86) I; daf-18(ok480) IV.

LRB429 akt-1(mg144gf) V; pdk-1(mg142gf) X.

LRB430 daf-18(ok480) IV; akt-1(mg306) V.

LRB441 age-1(m333) II; daf-18(ok480) IV.

LRB461 lin-35(n745) daf-16(mu86) I.

LRB486 lin-35(kea7[lin-35p::degron::GFP::lin-35]) I; keaSi10[rpl-28p::TIR1::mRuby::unc-54 3'UTR + Cbr-unc-119(+)] II; daf-18(ok480) IV.

LRB487 keaSi10 [rpl-28p::TIR1::mRuby::unc-54 3'UTR + Cbr-unc-119(+)] II.

LRB548 lin-35(kea7[lin-35p::degron::GFP::lin-35]) I.

LRB550 lin-35(kea7) I; daf-18(syb1659[daf-18::degron::3xFLAG] syb6062[D137A]) IV.

LRB565 +/qC1 [dpy-19(e1259) glp-1(q339)] nIs189 III.

LRB589 +/qC1 [dpy-19(e1259) glp-1(q339)] nIs189 III; daf-18(ok480) IV.

LRB595 clp-1(tm858) III; daf-18(ok480) IV.

LRB596 +/hT2 [bli-4(e937) let-?(q782) qIs48] (I;III)

LRB598 lin-35(kea7[lin-35p::degron::GFP::lin-35]) I; keaSi10[rpl-28p::TIR1::mRuby::unc-54 3'UTR + Cbr-unc-119(+)] II; clp-1(tm858) III; daf-18(ok480) IV.

LRB599 lin-35(kea7[lin-35p::degron::GFP::lin-35]) I; keaSi10[rpl-28p::TIR1::mRuby::unc-54 3'UTR + Cbr-unc-119(+)] II; clp-1(tm858) III.

LRB610 kuEx119[sur-5::GFP + C32F10]

LRB611 daf-18(ok480) IV; kuEx119[sur-5::GFP + C32F10]

LRB634 daf-18(ok480) IV; efl-1(se1) V.

LRB635 daf-18(ok480) IV; lin-15B(n744) X.

LRB638 lin-36(we36) III; daf-18(ok480) IV.

LRB652 lin-35(kea7[lin-35p::degron::GFP::lin-35 syb8778[K541A]) I; daf-18(ok480) IV.

LRB654 +/hT2 [bli-4(e937) let-?(q782) qIs48] (I;III); daf-18(ok480) IV.

LRB659 lin-35(kea7[lin-35p::degron::GFP::lin-35]) I; daf-18(ok480) IV.

LRB673 lin-9(n942)/qC1 [dpy-19(e1259) glp-1(q339)] nIs189 III.

LRB682 lin-9(n942)/qC1 [dpy-19(e1259) glp-1(q339)] nIs189 III; daf-18(ok480) IV.

LRB683 +/mIn1 [mIs14 dpy-10(e128)] II.

LRB685 lin-53(n3368) I/hT2 [bli-4(e937) let-?(q782) qIs48] (I;III); daf-18(ok480) IV.

## C. elegans maintenance

All strains assayed in this study were maintained with Escherichia coli (E. coli) OP50 on nematode growth medium (NGM) plates and were well fed for at least three generations before being used in experiments. Worms were cultured and starved at 20°C. Unless otherwise noted, all processes in Materials and Methods section were performed at 20°C.

## Auxin preparation and addition

A 400 mM indole-3-acetic acid (auxin) master stock was prepared in ethanol and stored in the dark at −20°C. A working stock of 133 mM auxin was prepared by diluting the master stock with ethanol, and it was also stored in the dark at −20°C. For experiments, the working stock was added to the culture at 200 µM just after isolating embryos by hypochlorite treatment, making it 0.15% ethanol. 0.15% ethanol without auxin was added to control cultures.

## Starvation survival

For each biological replicate and each strain, seven L4 worms were picked onto four 10 cm NGM plates seeded with OP50 (total: 28 L4 larvae). 96 h later, those large plates were hypochlorite treated to collect embryos from gravid adults (Hibshman et al, 2021). Those embryos were then resuspended, washed, counted, and cultured in either S-basal with 0.1% ethanol (for Figs 1, 3A, 5D, 6D, S3D, and S4B), S-basal with 0.15% ethanol or 200 µM auxin (for Fig 3B), or S-basal with 0.15% ethanol (for Fig 5E). Cultures had ~1 embryo/µl in 5 ml and were placed in 16 mm glass tubes at 20°C in the dark on a tissue-culture roller drum (New Brunswick TC-7) at ~30 rpm so embryos hatched and entered L1 arrest (Hibshman et al, 2021). The day after hypochlorite treatment, and again every day after that, a 100 µl aliquot of each starvation culture was plated on a 6 cm NGM plate around an E. coli OP50 lawn in the center. The number of larvae in the aliquot was recorded (total plated). Two days later, the number of live worms on the lawn was recorded (total alive). Proportion alive was determined as total alive divided by total plated. For all strains harboring lin-35(n745), efl-1(se1), lin-15B(n744), and the hT2[GFP] balancer, 10 L4 larvae were picked onto at least six 10 cm NGM plates (total: at least 60 L4s), and 120 h later, those large plates were hypochlorite treated. Everything else was the same as described above.

## Statistics for starvation survival

For each genotype in each replicate, proportion alive on each day was normalized to the first day of starvation. Survival curves were fit using quasi-binomial logistic regression, with the response variable being normalized proportion alive and the explanatory variable being days of starvation. Half-lives were calculated for each survival curve, and replicate half-lives were subjected to Bartlett's test to assess variance homogeneity across genotypes. Two-tailed unpaired t tests were performed on half-lives to compare genotypes, and variance was pooled if Bartlett's test suggested homogeneous variance.

## Hatching curve for RNA-seq

Gravid adult worms were hypochlorite treated to collect embryos (see above), which were then resuspended, washed, counted, and cultured in S-basal with 0.1% ethanol. Cultures had ~1 embryo/µl in

20 ml (~20,000 embryos per culture) and were placed in Erlenmeyer flasks at 20°C in a shaking incubator at 180 rpm, so embryos hatched and entered L1 arrest (Hibshman et al, 2021). Starting 12 h after hypochlorite treatment, a 100 $\mu$l aliquot was sampled from the culture every hour, and the numbers of hatched and unhatched embryos were recorded. Proportion hatched was calculated as the number of hatched embryos divided by the total number of sampled embryos. At 12 h after hypochlorite treatment, the average hatching efficiency (proportion hatched) for all cultures was about 50%, and at 16 h after hypochlorite treatment, all genotypes reached maximal hatching efficiency (Fig S1A). 16 h after hypochlorite treatment was chosen as the timepoint for RNA-seq sample collection.

### RNA-seq sample collection

Worms were hypochlorite treated and embryos were cultured as described in *Hatching curve for RNA-seq*. After 16 h of hypochlorite treatment, cultures were transferred to 15 ml conical tubes and spun at 3,000 rpm for 1 min. Starved larvae were transferred to 1.5 ml Eppendorf tubes with Pasteur pipets in 100 $\mu$l or less, and the tubes were snap-frozen in liquid nitrogen and stored at −80°C until RNA isolation. Three biological replicates were performed.

### RNA isolation and RNA-seq library preparation

RNA was extracted using TRIzol Reagent (#15596026; Invitrogen) using the manufacturer's protocol with some exceptions. 100 $\mu$L of acid-washed sand (#27439; Sigma-Aldrich) was added to each sample at the beginning of the extraction protocol to aid with homogenization. RNA was eluted in nuclease-free water and stored at −80°C until further use. Libraries were prepared for sequencing using the NEBNext Ultra II RNA Library Prep Kit for Illumina (#E7775; New England Biolabs) starting with 50 ng of total RNA per sample as input and 14 cycles of PCR. Barcoded libraries were pooled and sequenced on the NovaSeq 6000 S-Prime flow cell to obtain 50 bp paired-end reads. See the README sheet in Supplemental Data 1 for the number of reads obtained per library.

### Differential expression analysis of RNA-seq data

Bowtie2-2.3.3.1-linux-x86_64 (Langmead & Salzberg, 2012) was used to map paired-end reads with the command bowtie2 -p 2 -k 1 -S.1 -m 2 -S -p 2. The average mapping efficiency was 90.0%, and the SD was 1.5% (Supplemental Data 1). HTSeq python-htseq 0.6.1p1-4build1 (amd64 binary) in ubuntu bionic (Anders et al, 2015) was used to count reads against *C. elegans* genome version WS273. Count data were restricted to include only protein-coding genes (20,127). Before differential expression analysis, the gene list was further restricted to include only genes with counts per million (CPM) > 1 in at least three libraries (15,018 genes). Counts were then normalized using the Trimmed Mean of M-values method using edgeR 3.28.1 (Robinson et al, 2010). PCA in Fig S1B was performed on the set of 15,018 reproducibly detected genes using prcomp function in R stats package. The glmQLFit and glmQLFTest functions in edgeR found 871 DEG out of 15,018 in at least one genotype amongst *daf-16*, *daf-18*, *daf-16; daf-18*, and WT. For hierarchical clustering in Fig 2A, log$_2$(CPM) values for these 871 genes were averaged across

replicates within genotype, z-score normalized, and clustered using hclust function in R stats package. The exactTest function in edgeR was used to find genes differentially expressed between pairs of genotypes. CPM values for each gene, genotype, and replicate along with *P*-values for differential expression analysis are available in Supplemental Data 1.

### Transcriptome-wide epistasis analysis

The transcriptome-wide epistasis coefficient is described elsewhere (Angeles-Albores et al, 2018), but it is essentially the slope of a regression over all significant DEGs when the observed log$_2$FCs in the double mutant is plotted as a function of the expected log$_2$FCs in the double mutant based on log$_2$FCs in each single mutant assuming the two genes interact log-additively (i.e., function independently). Raw count values were processed and genes were filtered as described in *Differential expression analysis of RNA-seq data* (restricted to protein-coding genes and CPM > 1 in at least 3 libraries), resulting in 15,018 genes. Count normalization and differential expression analysis of each mutant-versus-WT was conducted using DESeq2 1.30.1 (Love et al, 2014). DESeq2 results were used in a transcriptome-wide epistasis analysis pipeline as described (Angeles-Albores et al, 2018). Specifically, DESeq2 generated gene-wise log$_2$FCs for *daf-16*, *daf-18*, and *daf-16; daf-18* compared with WT, and standard errors and q-values of those log$_2$FCs. Q-value < 0.1 was used as the cutoff to extract DEGs in mutants compared with WT. The transcriptome-wide epistasis analysis pipeline (Angeles-Albores et al, 2018) used DEGs shared by all three mutant-versus-WT comparisons (563 genes) to fit predefined models and a parameter-free model. The predefined models are summarized in Fig 2B. The parameter-free model does not have underlying presumptions. Log$_2$FCs for the 563 shared DEGs were bootstrapped 5,000 times (choosing 563 genes with replacement) to fit each model, and the distribution of transcriptome-wide epistasis coefficients was plotted (Fig 2C). Besides generating the distribution of transcriptome-wide epistasis coefficients, model-fitting also generated a likelihood for each model using Bayesian statistics (Angeles-Albores et al, 2018). Odds ratio (OR) was computed by dividing the model likelihood of the parameter-free model by each predefined model. OR > 10$^3$ was used as the cutoff to reject predefined models.

### GSEA

*daf-16*-independent targets of *daf-18* (as defined in Fig 2D) were analyzed with WormExp v2.0 (Yang et al, 2016), a web application that tests if a user's input gene set is enriched for other experimental gene sets using Fisher's Exact test statistics. WormExp's default FDR < 0.1 was used as the cutoff to assess significant enrichment. The results and statistics are in Supplemental Data 2.

Hypergeometric tests were conducted for Figs 2D, E, and G, 6B, S1C and D, and S4A. In those Venn-diagram style plots, hypergeometric tests were performed by comparing two gene sets in the Venn diagram, with the background set being all genes detected by RNA-seq (n = 15,018). The other type of plot was generated using UpSetR 1.4.0 (Conway et al, 2017), which displays the number of genes shared by different combinations of multiple gene sets.

Hypergeometric tests were performed by comparing *daf-16*-independent targets of *daf-18* (Fig 6B) or *daf-18*-independent targets of *daf-16* (Fig S4A) to the other gene sets in the plot.

The Kolmogorov-Smirnov tests were used to assess the equality between cumulative distributions in Figs 2F and H and 6B and C. All comparisons were against "all detected genes."

### Western blot sample collection

Samples were collected the same way as described in *RNA-seq sample collection* except that starvation media were either virgin S-basal (no ethanol or cholesterol) with 0.15% ethanol or 200 *μM* auxin, and protease inhibitors (4693159001; Millipore Sigma) were included. Frozen samples were freeze-thawed three times, using liquid nitrogen and a 45°C water bath. Then, sample buffer (S3401; Millipore Sigma) was added, and samples were boiled for 10 min at 95°C, frozen for 15 min on dry ice, boiled again for 10 min at 95°C, and centrifuged at 14,000*g* for 10 min to pellet any worm debris. The supernatant was collected, used to measure protein concentrations using the Pierce 660 nm Protein Assay (22662; Thermo Fisher Scientific), and used in Western blots. Three biological replicates were performed. Fed L1 samples were prepared following the same protocol except that embryos from hypochlorite treatment were grown in S-complete with 25 mg/ml *E. coli* HB101 and 0.15% ethanol and that samples were briefly washed three times in virgin S-basal with 0.15% EtOH to eliminate bacterial food right before sample collection.

### Western blot

~2 *μg* of protein was loaded per lane on a 4–12% Bis-Tris gel (NP0321BOX; Thermo Fisher Scientific). The gel was run at 200 V for 50 min and was then transferred at 30 V for 1 h to a polyvinylidene fluoride (PVDF) membrane (LC2005; Thermo Fisher Scientific). The membrane was blocked for 1 h using 3% milk (1706404; Bio-Rad) dissolved in 1x Tris-Buffered Saline with Tween 20 (TBST), thoroughly washed three times using 1x TBST, and then incubated overnight at 4°C in 1:5,000 1x TBST-diluted primary antibody against GFP (sc-9996; Santa Cruz). The membrane was then thoroughly washed three times using 1x TBST and incubated for 1 h in 1:10,000 1x TBST-diluted HRP-conjugated secondary antibody against Mouse IgG (115-035-166; Jackson ImmunoResearch). The membrane was then thoroughly washed three times using 1x TBST and then blotted using a chemiluminescent assay (34094; Thermo Fisher Scientific). Western blot images were acquired when a single-dot saturation was seen in any band on the membrane. Band intensity was quantified using ImageJ 1.54f (Schindelin et al, 2012) with background intensity subtracted. Western blot images and quantification are in Figs 4A and B and 5B and C.

### Immunoprecipitation (IP) and mass spectrometry (MS) sample collection

~6 million starved L1s were collected following the protocol in Hibshman et al [2021]) with minor modifications. Specifically, gravid worms raised on plates were hypochlorite-treated to obtain embryos, and embryos were cultured in liquid at 20°C, 180 rpm, at a density of 5 worm/*μl*, and with 50 mg/ml *E. coli* HB101 until they were gravid adults (72 h). Adults were washed and hypochlorite-treated to obtain millions of embryos. Those embryos were cultured in virgin S-basal with 0.1% ethanol at 5 embryos/*μl* in 2.8 liters Erlenmeyer flasks in a shaking incubator at 180 rpm, so embryos hatched and entered L1 arrest (Hibshman et al, 2021). 16 h later, those cultures were washed and collected in IP Buffer (50 mM Tris-Cl pH 7.5, 100 mM KCl, 2.5 mM MgCl$_2$, 0.1% NP-40 Alternative [492016; Millipore Sigma]), which was precooled to 4°C and contained phosphatase inhibitors (4906845001; Millipore Sigma) and protease inhibitors (4693159001; Millipore Sigma). At this point, samples looked slurry-like. The worm slurries were frozen into "popcorn" by dripping them into liquid nitrogen. Worm popcorn was stored at –80°C until sample processing. All sample processing was performed at 4°C. Worm popcorn was homogenized using mortar and pestles that were precooled using liquid nitrogen, and the homogenate was centrifuged at 16,000*g* for 10 min to remove insoluble material. The supernatant was the "input sample." 80 *μl* of the input sample was used to measure protein concentrations, as described in *Western blot sample collection*. The remainder of the input sample was then diluted to 1 *μg/μl* with IP buffer and was used for IP.

### Immunoprecipitation (IP) and mass spectrometry (MS)

The entire IP process was performed at 4°C. After diluting input samples to 1 *μg/μl* as described in *Immunoprecipitation (IP) and mass spectrometry (MS) sample collection*, 20 *μl* of anti-GFP bead slurry (ChromoTek gta) was added to each sample. The samples were incubated on a nutator for 1 h. Then, beads were pelleted by centrifuging at 500*g* for 30 s. Beads were then washed thoroughly eight times by adding IP buffer, incubating on a nutator, pelleting beads at 500*g* for 30 s, and removing the supernatant. The total time of eight washes was ~30 min. 70 *μl* of each sample was submitted to Duke Proteomics Core for MS analysis, and the remainder was used to determine sample concentrations and Western blot (Fig 4C and D), as described in *Western blot sample collection* and *Western blot*, respectively. Quantitative LC–MS/MS was performed using an MClass UPLC system (Waters Corp) coupled to a Thermo Orbitrap Fusion Lumos high-resolution accurate mass tandem Mass Spectrometer (Thermo Fisher Scientific) equipped with a FAIMSPro device via a nanoelectrospray ionization source. For additional details regarding LC–MS/MS contact the corresponding author.

## Data Availability

RNA-seq data: Gene Expression Omnibus GSE281157. Code to reproduce all results presented in this paper: https://github.com/jc271828/daf18_and_lin35.

## Supplementary Information

# Acknowledgements

We would like to thank Aaron Reinke and David Fay for sharing strains. We would like to thank Shihui Chen for sharing IP/MS protocols. Some strains were provided by the CGC, which is funded by NIH Office of Research Infrastructure Programs (P40 OD010440). We would also like to thank WormBase. This work was funded by the National Institutes of Health (R01GM117408 and R01GM143159, awarded to LR Baugh).

## Author Contributions

J Chen: conceptualization, data curation, formal analysis, validation, investigation, visualization, methodology, and writing—original draft, review, and editing.
R Chitrakar: data curation.
LR Baugh: conceptualization, resources, supervision, funding acquisition, methodology, project administration, and writing—original draft, review, and editing.

## Conflict of Interest Statement

The authors declare that they have no conflict of interest.

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
