## [Reviewer comments · Life Science Alliance]

Life Science Alliance

DAF-18/PTEN protects LIN-35/Rb from CLP-1/CAPN-mediated cleavage to promote starvation resistance

Jingxian Chen, Rojin Chitrakar, and L. Ryan Baugh

DOI: <https://doi.org/10.26508/lsa.202403147>

Corresponding author(s): L. Ryan Baugh, Duke University

Review Timeline:

Submission Date:	2024-11-25
Editorial Decision:	2025-01-06
Revision Received:	2025-03-24
Editorial Decision:	2025-03-26
Revision Received:	2025-03-26
Accepted:	2025-03-27

Transaction Report:

January 6, 2025

Re: Life Science Alliance manuscript #LSA-2024-03147-T

Ryan Baugh
Duke University
Department of Biology
Box 90338
Durham, NC x27708

Dear Dr. Baugh,

Thank you for submitting your manuscript entitled "DAF-18/PTEN protects LIN-35/Rb from CLP-1/CAPN-mediated cleavage to promote starvation resistance" to Life Science Alliance. The manuscript was assessed by expert reviewers, whose comments are appended to this letter. We invite you to submit a revised manuscript addressing the Reviewer comments.

Thank you for this interesting contribution to Life Science Alliance. We are looking forward to receiving your revised manuscript.

Sincerely,

B. MANUSCRIPT ORGANIZATION AND FORMATTING:

Reviewer #1 (Comments to the Authors (Required)):

In this study, Chen et al. investigate starvation resistance in *C. elegans* as mediated by the tumor suppressor PTEN/DAF-18, a protein known to antagonize insulin signaling and activate DAF-16/FoxO. The authors demonstrate that DAF-18's role in starvation resistance is independent of AGE-1 and DAF-16, proposing that DAF-18's protein phosphatase activity protects LIN-35/Rb from cleavage by CLP-1. This work fills an important gap in understanding the DAF-16-independent functions of DAF-18, and the authors' expertise in assessing starvation resistance in L1 nematodes provides a valuable framework. While the study employs rigorous genetic approaches, some conclusions are weakly supported or overstated. Significant revisions are needed to improve clarity and address overstated claims, particularly in the lengthy discussion section.

Revisions and Suggestions:

1. Use of Standard Genetic Terminology:

o In line 142, the phrase "completely abolish" should be replaced with "suppress".

o Similarly, in the same paragraph, the assumptions regarding the ability of GoF alleles to phenocopy *daf-18*(null) starvation sensitivity are overstated and should be moderated.

2. Precision in Wording:

o Line 160: Replace "does nothing" with "fails to suppress starvation sensitivity in a *daf-18* null background."

o Line 161: Clarify that a two-way ANOVA is a statistical test for significance and not evidence of an interaction between two mutations.

3. Experimental Clarity:

o In the paragraph starting at line 168, the methods used to synchronize L1s could be simplified for clarity. The discussion of potential indirect effects is currently unclear.

o Line 179: Revise to accurately describe that PC1 is the genotype, and do the same for line 185.

4. Statistical Analysis:

o Lines 202-211 require clarification to provide better context for the statistical analysis. Currently, this section reads as a list of statistical terms with insufficient explanation.

o At line 222, include statistical data for the S2 dataset. Additionally, revise the phrase "unexpected but not surprising," which is contradictory.

5. Genetic Pathway Discussion:

o The discussion in the paragraph ending at line 245 requires refinement. Demonstrating shared targets between two mutants may suggest involvement in the same pathway but does not distinguish between linear and parallel pathways.

o The inviability of the *lin-35; daf-18* double mutant (line 257) seems to contradict a linear pathway model and should be addressed.

o Line 252: Like line 161, clarify that an ANOVA test cannot indicate the presence or absence of a genetic interaction.

6. Methodological Context:

o Lines 250-266: Include details on the timing of auxin addition in the LIN-35::AID experiments to clarify whether this represents post-developmental depletion.

7. Figure Clarity:

o Line 306: The enrichment of N-terminal versus C-terminal peptides in S2 is difficult to discern. If the difference is subtle, provide statistical support.

o Line 317: Consider whether "increased" should be replaced with "decreased" based on the data.

8. Discussion Section:

o The discussion feels repetitive and should be significantly condensed. Sentences from lines 472-477, in particular, need editing for clarity. Terms like "non-selective disruption" and "these alleles" should be defined.

o Line 459: The results presented do not directly demonstrate DAF-18's protein phosphatase activity in promoting starvation resistance. This is acknowledged later in the paragraph but should be addressed more consistently. Avoid speculative statements such as "we believe" (line 466).

Reviewer #2 (Comments to the Authors (Required)):

This manuscript by Chen et al, titled "DAF-18/PTEN protects LIN-35/Retinoblastoma(Rb) from CLP-1/CAPN-mediated cleavage

to promote starvation resistance", explores a non-canonical output of DAF-18/PTEN in the fascinating process of survival of starvation by the *C. elegans* first larval stage, L1. The manuscript is an excellent ride: a) DAF-18 promotes starvation resistance both dependently and independently of IIS (insulin/insulin-like growth factor signaling), and hence has outputs dependent and independent upon DAF-16/FOXO. b) This signal is possibly through protein (not lipid) phosphatase activity, c) DAF-18 prevents cleavage and destabilization of LIN-35/Rb by CAPN-1/calpain, and d) LIN-35 functions in conjunction with the DREAM complex to mediate resistance to starvation. This model is derived from a series of elegant genetic interactions using animal assays of starvation survival of single and double mutant as well as transcriptomics epistasis using readouts of RNAseq and comparisons of lists of DEGs derived in this study vs other studies. The conclusions are sound and are supported by the data.

The study provides intriguing mechanistic insights into both starvation response and the interrelationship of tumor suppressors PTEN and RB.

The mechanism by which CAPN-1/Calpain is regulated by the IIS-independent function of DAF-18/PTEN is not shown. This will be the interesting subject of a future study. Also of interest in the future would be development of transcriptional reporters of this signal-gene regulatory network for screening purposes.

The manuscript is also well written. Minor recommendations follow below.

Major:

1. The transcriptome epistasis would benefit greatly from a schematic to unpack the complex relationships. As is, I had to re-read a few times to understand the point. But still just a graphical presentation issue.: no experiments

Minor comments

- Ln 83: dauer formation, first mention of it. Define it in the context of diapause above?
- Ln 89: Does SynMuv role of lin-35 matter?
- 89: "and express germline genes in the soma" is vague. I think you mean repress, like mammalian RB with E2F, and the DREAM complex, directly below
- Fig 1: these panels would benefit from lateral expansion to assist discrimination of different genotypes, especially for 1C. Maybe stack them vertically so they are presented as one column width?
- 136-138: Should probably add "with the caveat that m333 homozygotes could theoretically be rescued by maternal gene product from m333/+ mothers and thus perhaps not be null for promoting starvation-dependent lethality" or some such. (This minor concern is offset by subsequent survival genetics)
- 141: should probably reference the two Paradis and Ruvkun paper rather than the Murphy Workbook chapter: maybe where the 1999 Paradis ref is below. Also, please use nomenclature "pdk-1(mg142gf)" and same for akt-1 in text
- Fig 1B: Dk blue and black lines difficult to distinguish. Maybe make the double mutant bright purple, like used in 1A?
- 160: change "showing" to "suggesting"
- 172: "We wanted to capture relatively early, direct effects of daf-16 and daf-18 rather than indirect effects." But can one ever say that for a transcriptome? Change expression of a TF and you get knock on effects that are also indirect. I would say "complementary"
- 196: are>were
- As with Fig 1, Fig 3B would benefit from spreading it out laterally so the reader can see the different K-M curves. Maybe A atop B?
- 272: daf-18 (+)
- Fig 5C: Align with lanes in Fig 5B for easier understanding

For future reference we recommend switching to tmC# balancers for transcriptomic analyses; they have far fewer background mutations.

Reviewer #3 (Comments to the Authors (Required)):

In Chen et al. the authors describe detailed epistasis and gene expression experiments that nicely lay out a new role for DAF-18/PTEN in L1 arrest in *C. elegans*. The authors clearly describe why their data support a model where DAF-18/PTEN acts as a protein-phosphatase to regulate LIN-35/Rb, which in turn plays a role in regulating L1 arrest. This is a newly identified role for DAF-18/PTEN which has only been studied in *C. elegans* as a lipid phosphatase up to this point. In addition, the regulation of LIN-35 by DAF-18 has not previously been found. These novel findings are additionally exciting because of the conserved nature of both of these proteins, which are each known to be important tumor suppressors. Overall, the experiments are sound and logically laid out. Minor changes in writing are needed to make the dual roles of DAF-18/PTEN as both a protein and lipid phosphatase both being important for L1 diapause need to be clarified and there needs to be some further discussion of what the potential role of germline gene expression may be in this phenotype.

Specific Comments:

Major Comments:

1) While the authors make clear in abstract and the summary figure in Fig 7 that their model is that DAF-18/PTEN plays a dual role as both a lipid and protein phosphatase in L1 arrest, during the majority of the Results section they focus exclusively on the role of DAF-18/PTEN as a protein-phosphatase such that they don't fully explain some of their data, which highlights this dual role.

For example: Figure 1A and B: The authors show that *daf-18* mutants have significant loss of survival of L1 arrest compared to WT. In addition, they show that *daf-18; age-1* double mutants are also significantly less likely to survive L1 arrest compared to WT. However, the double mutant is not the same as the single *daf-18* mutant and shows some partial suppression of the decreased survival of the L1 arrest. The authors do not give data to indicate if in fact the partial suppression is significant, but it certainly looks significant. Given this, it seems that this is an example of the two different aspects of PTEN function playing a role in starvation arrest- the first being dephosphorylation of lipids and the second being dephosphorylation of proteins. The data given in Fig 1B supports the idea that there is some aspect of PTEN signaling with lipids as the substrate being important in L1 arrest survival. It would be good for the authors to acknowledge this in the written explanation of Fig 1.

Similarly in places throughout the written text where the data points towards the dual role of DAF-18/PTEN should be explicitly acknowledged.

2) The authors propose a model where "DREAM represses germline gene expression in the germline and soma downstream of DAF-18/PTEN during L1 arrest". However, this doesn't take into account that the DREAM complex (including LIN-35 which is a member of the DREAM complex) together with LIN-15B repress germline gene expression in the soma normally starting in the embryonic stage. This is demonstrated by the beginning of ectopic germline gene expression in embryos in DREAM complex mutants (Petrella 2011). The author's discussion of their pathway doesn't acknowledge this and makes it seem like the DREAM complex/DAF-18/LIN-15B are only necessary during L1 arrest for this action. Is it possible that DAF-18 is necessary for the repression of germline genes generally, including during late embryogenesis? Or do the authors think the proposed protein phosphatase activity is only necessary during L1 arrest? Would their model change at all if it was the former or how would they take into account the embryonic activity of DREAM/LIN-15B?

3) Figure 5B and C: While it is clear that there is no significant decrease of LIN-35 protein intensity in the *clp-1;daf-18* double compared with the *clp-1* single- it does seem that there is a significant decrease in LIN-35 protein intensity in both the *clp-1* mutant strains compared to WT. If CLP-1 is function to degrade LIN-35 through cleavage- shouldn't there be an increase in LIN-35 protein levels not a decrease when compared to WT? The authors need to explain this discrepancy in their model.

Minor Comments

The authors are investigating a lot of different proteins in related signaling pathways. While the pathway laid out in Fig 7 as the summary of the paper is great, a diagram of those pathways and what is being investigated would be really helpful in Fig 1 for readers to keep track of the different parts the authors are investigating.

Lines 101-102: The author indicate that mutations in one MuvB core member *lin-37* result in ectopic expression of germline genes in the soma. However, loss of any of the five MuvB core proteins and also the DP homolog *dpl-1* result in ectopic expression of germline genes (Petrella 2011, Wu et al 2012). The way it is written it suggests that only loss of *lin-37* shows this result, which is inaccurate.

Figure 1C: It is really hard to see the double mutant with the same color as the single *daf-16* mutant- I suggest choosing a different contrasting color for the double mutant.

Small suggested change Line 453-54: Original "though protein-phosphatase activity has not been demonstrated" change to "though protein-phosphatase activity is yet to be shown in *C. elegans*." Since the argument being made is that it does have this activity in *C. elegans* and the data presented suggests that- this is more compelling.

We would like to thank the reviewers for their time and effort. The reviews were uniformly constructive, and the manuscript is improved in revision. We provide a point-by-point response to each reviewer comment below, with our responses in blue font.

Reviewer #1 (Comments to the Authors (Required)):

In this study, Chen et al. investigate starvation resistance in *C. elegans* as mediated by the tumor suppressor PTEN/DAF-18, a protein known to antagonize insulin signaling and activate DAF-16/FoxO. The authors demonstrate that DAF-18's role in starvation resistance is independent of AGE-1 and DAF-16, proposing that DAF-18's protein phosphatase activity protects LIN-35/Rb from cleavage by CLP-1. This work fills an important gap in understanding the DAF-16-independent functions of DAF-18, and the authors' expertise in assessing starvation resistance in L1 nematodes provides a valuable framework. While the study employs rigorous genetic approaches, some conclusions are weakly supported or overstated. Significant revisions are needed to improve clarity and address overstated claims, particularly in the lengthy discussion section.

Thank you for carefully reviewing our manuscript, and for noting its merits as well as opportunities to improve it.

Revisions and Suggestions:

1. Use of Standard Genetic Terminology:

o In line 142, the phrase "completely abolish" should be replaced with "suppress".

Done

o Similarly, in the same paragraph, the assumptions regarding the ability of GoF alleles to phenocopy *daf-18*(null) starvation sensitivity are overstated and should be moderated.

We agree that we did not originally present the limitation of this approach clearly enough. The text did include the statement "then *sufficiently* increasing AGE-1/PI3K signaling activity should phenocopy *daf-18* null mutants", but we revised the final sentence of the paragraph to make the caveat more clear: "These results could be due to inadequate activation of AGE-1/PI3K signaling with these gain-of-function alleles, but they are also consistent with DAF-18/PTEN promoting starvation resistance independently of AGE-1/PI3K signaling..."

2. Precision in Wording:

o Line 160: Replace "does nothing" with "fails to suppress starvation sensitivity in a *daf-18* null background."

Done

o Line 161: Clarify that a two-way ANOVA is a statistical test for significance and not evidence of an interaction between two mutations.

We now specify that the interaction is "statistical"

3. Experimental Clarity:

o In the paragraph starting at line 168, the methods used to synchronize L1s could be simplified for clarity. The discussion of potential indirect effects is currently unclear.

We have revised the sentences regarding sample collection for clarity, comments about direct/indirect effects have been deleted.

o Line 179: Revise to accurately describe that PC1 is the genotype, and do the same for line 185.

Thank you for the suggestion, but it is not appropriate to say that PC1 “is” the genotype since the factors in the experimental design are not modeled in PCA and could be captured by one or more components. The mutants actually do not fully separate from wild type in PC1 but instead separate in the plane defined by PC1 and PC2. We revised the text to make this more explicit.

4. Statistical Analysis:

o Lines 202-211 require clarification to provide better context for the statistical analysis. Currently, this section reads as a list of statistical terms with insufficient explanation. We have revised this passage to help readers without adequate understanding of statistics understand the gist of the approach. We have also added a graphic to better convey the logic of the alternative models to the reader, as suggested by Reviewer 3. We recognize that this approach is relatively novel and sophisticated, and that it requires adequate technical expertise in statistics to fully comprehend, but it is a powerful approach that strengthens the RNAseq analysis, so we included it. We have done our best to make it accessible to the reader in the cited Results section as well as the Methods. We feel like it would be a distraction to more completely explain this approach, and we think it would be better for the reader to refer to the Methods and cited paper if additional explanation is needed.

o At line 222, include statistical data for the S2 dataset. Additionally, revise the phrase “unexpected but not surprising,” which is contradictory.
Done and done.

5. Genetic Pathway Discussion:

o The discussion in the paragraph ending at line 245 requires refinement. Demonstrating shared targets between two mutants may suggest involvement in the same pathway but does not distinguish between linear and parallel pathways.
Done

o The inviability of the *lin-35*; *daf-18* double mutant (line 257) seems to contradict a linear pathway model and should be addressed.
Good point, thank you. We now specify that it is inviable “without being starved” where this statement is made in the Results. We now circumscribe the conclusion about them comprising a linear pathway to L1 starvation resistance in the Discussion, and we have added a sentence interpreting the inviability of the double mutant: “Notably, the *lin-35*; *daf-18* double null mutant was inviable, despite being well fed, suggesting independent function of these two pleiotropic genes in at least one other process impacting viability.”

o Line 252: Like line 161, clarify that an ANOVA test cannot indicate the presence or absence of a genetic interaction.
Done

6. Methodological Context:

o Lines 250-266: Include details on the timing of auxin addition in the LIN-35::AID experiments to clarify whether this represents post-developmental depletion.
Done.

7. Figure Clarity:

o Line 306: The enrichment of N-terminal versus C-terminal peptides in S2 is difficult to discern. If the difference is subtle, provide statistical support.

It is true that the difference in Fig S2B is difficult to discern, but S2C provides an alternative, simplified way of plotting the data that makes the difference easier to discern, and it provides statistical support.

o Line 317: Consider whether "increased" should be replaced with "decreased" based on the data.

Good catch; we fixed this.

8. Discussion Section:

o The discussion feels repetitive and should be significantly condensed. Sentences from lines 472-477, in particular, need editing for clarity. Terms like "non-selective disruption" and "these alleles" should be defined.

This passage has been revised for clarity, and the indicated terms have been defined. It was redundant with a passage in the Intro, and we decided to remove this from the Intro but keep the version in the Discussion. We have also been through the entire Discussion, removing words and sentences that are not necessary.

o Line 459: The results presented do not directly demonstrate DAF-18's protein phosphatase activity in promoting starvation resistance. This is acknowledged later in the paragraph but should be addressed more consistently. Avoid speculative statements such as "we believe" (line 466).

We have toned down our language, included alternative possibilities earlier in the paragraph, and included additional alternative possibilities.

Reviewer #2 (Comments to the Authors (Required)):

This manuscript by Chen et al, titled "DAF-18/PTEN protects LIN-35/Retinoblastoma(Rb) from CLP-1/CAPN-mediated cleavage to promote starvation resistance", explores a non-canonical output of DAF-18/PTEN in the fascinating process of survival of starvation by the *C. elegans* first larval stage, L1. The manuscript is an excellent ride: a) DAF-18 promotes starvation resistance both dependently and independently of IIS (insulin/insulin-like growth factor signaling), and hence has outputs dependent and independent upon DAF-16/FOXO. b) This signal is possibly through protein (not lipid) phosphatase activity, c) DAF-18 prevents cleavage and destabilization of LIN-35/Rb by CAPN-1/calpain, and d) LIN-35 functions in conjunction with the DREAM complex to mediate resistance to starvation. This model is derived from a series of elegant genetic interactions using animal assays of starvation survival of single and double mutant as well as transcriptomics epistasis using readouts of RNAseq and comparisons of lists of DEGs derived in this study vs other studies. The conclusions are sound and are supported by the data.

The study provides intriguing mechanistic insights into both starvation response and the interrelationship of tumor suppressors PTEN and RB.

The mechanism by which CAPN-1/Calpain is regulated by the IIS-independent function of DAF-18/PTEN is not shown. This will be the interesting subject of a future study. Also of interest in the future would be development of transcriptional reporters of this signal-gene regulatory network for screening purposes.

The manuscript is also well written. Minor recommendations follow below.

Thank you highlighting the strengths of our manuscript and pointing out recommendations for improvement.

Major:

1. The transcriptome epistasis would benefit greatly from a schematic to unpack the complex relationships. As is, I had to re-read a few times to understand the point. But still just a graphical presentation issue.: no experiments

We added a schematic to Fig 2B to summarize the predefined models used in the analysis. We also revised the text in Results describing the analysis to improve clarity.

Minor comments

• Ln 83: dauer formation, first mention of it. Define it in the context of diapause above? \ Done.

• Ln 89: Does SynMuv role of lin-35 matter?

We think so given that other SynMuv genes come into play in the context of the DREAM complex. We left the reference to the SynMuv function of lin-35.

• 89: "and express germline genes in the soma" is vague. I think you mean repress, like mammalian RB with E2F, and the DREAM complex, directly below

We were referring to the mutant, but we have revised it for clarity. We now refer to lin-35/Rb function, and we mention lin-35 repressing expression of germline genes in the soma.

• Fig 1: these panels would benefit from lateral expansion to assist discrimination of different genotypes, especially for 1C. Maybe stack them vertically so they are presented as one column width?

Thank you for the suggestion, we have done this here and elsewhere as suggested, and it definitely improves the presentation of these critical survival results.

• 136-138: Should probably add "with the caveat that m333 homozygotes could theoretically be rescued by maternal gene product from m333/+ mothers and thus perhaps not be null for promoting starvation-dependent lethality" or some such. (This minor concern is offset by subsequent survival genetics)

Since daf-18(ok480) suppresses the maternal Daf-c phenotype of age-(m333) we did not use a balancer chromosome and the mothers were also homozygous double mutants. We added a sentence to spell this out to the reader.

• 141: should probably reference the two Paradis and Ruvkun paper rather than the Murphy Workbook chapter: maybe where the 1999 Paradis ref is below. Also, please use nomenclature "pdk-1(mg142gf)" and same for akt-1 in text

Done and done.

• Fig 1B: Dk blue and black lines difficult to distinguish. Maybe make the double mutant bright purple, like used in 1A?

We changed the colors to make them easier to distinguish.

• 160: change "showing" to "suggesting"

Done.

• 172: "We wanted to capture relatively early, direct effects of daf-16 and daf-18 rather than

indirect effects." But can one ever say that for a transcriptome? Change expression of a TF and you get knock on effects that are also indirect. I would say "complementary"

Given Reviewer 1's comments on this in addition to yours, we deleted this sentence.

- 196: are>were

We fixed this and the other incorrect tense in the first clause of the same sentence.

- As with Fig 1, Fig 3B would benefit from spreading it out laterally so the reader can see the different K-M curves. Maybe A atop B?

Yes, done.

- 272: daf-18 (+)

Done.

- Fig 5C: Align with lanes in Fig 5B for easier understanding

Thanks again, done.

For future reference we recommend switching to tmC# balancers for transcriptomic analyses; they have far fewer background mutations.

Thank you for the suggestion. We did not use balancers with our transcriptomic analysis, but the tmC# balancers are presumably better for all types of analysis.

Reviewer #3 (Comments to the Authors (Required)):

In Chen et al. the authors describe detailed epistasis and gene expression experiments that nicely lay out a new role for DAF-18/PTEN in L1 arrest in *C. elegans*. The authors clearly describe why their data support a model where DAF-18/PTEN acts as a protein-phosphatase to regulate LIN-35/Rb, which in turn plays a role in regulating L1 arrest. This is a newly identified role for DAF-18/PTEN which has only been studied in *C. elegans* as a lipid phosphatase up to this point. In addition, the regulation of LIN-35 by DAF-18 has not previously been found. These novel findings are additionally exciting because of the conserved nature of both of these proteins, which are each known to be important tumor suppressors. Overall, the experiments are sound and logically laid out. Minor changes in writing are needed to make the dual roles of DAF-18/PTEN as both a protein and lipid phosphatase both being important for L1 diapause need to be clarified and there needs to be some further discussion of what the potential role of germline gene expression may be in this phenotype.

Thank you for summarizing the significance of our work and for suggesting ways to improve it.

Specific Comments:

Major Comments:

1) While the authors make clear in abstract and the summary figure in Fig 7 that their model is that DAF-18/PTEN plays a dual role as both a lipid and protein phosphatase in L1 arrest, during the majority of the Results section they focus exclusively on the role of DAF-18/PTEN as a protein-phosphatase such that they don't fully explain some of their data, which highlights this dual role.

For example: Figure 1A and B: The authors show that *daf-18* mutants have significant loss of survival of L1 arrest compared to WT. In addition, they show that *daf-18; age-1* double mutants are also significantly less likely to survive L1 arrest compared to WT. However, the double mutant is not the same as the single *daf-18* mutant and shows some partial suppression of the decreased survival of the L1 arrest. The authors do not give data to indicate if in fact the partial suppression is significant, but it certainly looks significant. Given this, it seems that this is an example of the two different aspects of PTEN function playing a role in starvation arrest- the first being dephosphorylation of lipids and the second being dephosphorylation of proteins. The data given in Fig 1B supports the idea that there is some aspect of PTEN signaling with lipids as the substrate being important in L1 arrest survival. It would be good for the authors to acknowledge this in the written explanation of Fig 1.

Similarly in places throughout the written text where the data points towards the dual role of DAF-18/PTEN should be explicitly acknowledged.

Thank you for this suggestion. We were focused on defining a novel role of the putative protein-phosphatase activity of DAF-18, and we took the well-known role of its lipid-phosphatase activity for granted. We have added symbols for statistical significance of pairwise comparisons between genotypes addressing the lipid-phosphatase activity, and we have revised the text to explicitly address each of these results and state that they reflect the role of the known lipid-phosphatase activity. This includes Fig 1A&C, Fig 2 and Fig 3B.

2) The authors propose a model where "DREAM represses germline gene expression in the germline and soma downstream of DAF-18/PTEN during L1 arrest". However, this doesn't take into account that the DREAM complex (including LIN-35 which is a member of the DREAM complex) together with LIN-15B repress germline gene expression in the soma normally starting in the embryonic stage. This is demonstrated by the beginning of ectopic germline gene expression in embryos in DREAM complex mutants (Petrella 2011). The author's discussion of their pathway doesn't acknowledge this and makes it seem like the DREAM complex/DAF-18/LIN-15B are only necessary during L1 arrest for this action. Is it possible that DAF-18 is necessary for the repression of germline genes generally, including during late embryogenesis? Or do the authors think the proposed protein phosphatase activity is only necessary during L1 arrest? Would their model change at all if it was the former or how would they take into account the embryonic activity of DREAM/LIN-15B?

Thank you for pointing this out. In addition to ectopic germline expression being evident in embryos and fed larvae, *daf-18*'s effect on germline gene expression and primordial germ cell divisions also extends to fed larvae and potentially late embryos (Fry 2021). However, we found that LIN-35 abundance does not decrease in fed L1 larvae of a *daf-18* mutant, suggesting the pathway we have defined is specific to starvation. We have added these western blots results in a new Sup Fig (S2 Fig) that we cite in the Results section after presenting the results for L1 arrest (Fig 4). We have also added a paragraph to the end of the Discussion that raises the possibility that DAF-18 regulates LIN-35/DREAM in fed larvae, reviews the literature in support of this possibility (Petrella 2011; Fry 2021), but then mentions our negative western result in fed larvae, and concludes that regulation of LIN-35/DREAM by DAF-18 is restricted to starvation.

3) Figure 5B and C: While it is clear that there is no significant decrease of LIN-35 protein intensity in the *clp-1;daf-18* double compared with the *clp-1* single- it does seem that there is a significant decrease in LIN-35 protein intensity in both the *clp-1* mutant strains compared to WT.

If CLP-1 is function to degrade LIN-35 through cleavage- shouldn't there be an increase in LIN-35 protein levels not a decrease when compared to WT? The authors need to explain this discrepancy in their model.

Yes, this is an intriguing observation in want of an explanation. We appreciate that you call it out while still recognizing the key point that *clp-1* is required for loss of *daf-18* to cause LIN-35 cleavage. We agree that if anything, we would have expected LIN-35 abundance to increase in the *clp-1* mutant based on our model. It is possible that loss of *clp-1* has indirect effects on LIN-35 abundance stemming from unrelated mechanisms, such as transcription or translation. Alternatively, if CLP-1 is a direct target of DAF-18 for dephosphorylation, then perhaps CLP-1 protects LIN-35 from cleavage by one of the other six Calpain homologs when it is not phosphorylated, and it becomes proteolytically active when phosphorylated, as in a *daf-18* mutant. We chose to include the result in the interest of transparency, but we did not offer an explanation in the manuscript since it felt too speculative. Given your comment, we now call attention to it as an unexpected result, but we have elected to offer our speculation here and not in the manuscript.

Minor Comments

The authors are investigating a lot of different proteins in related signaling pathways. While the pathway laid out in Fig 7 as the summary of the paper is great, a diagram of those pathways and what is being investigated would be really helpful in Fig 1 for readers to keep track of the different parts the authors are investigating.

We agree that this will help many readers, and we have added a pathway diagram to Fig 1A to establish the research question and set the stage for what follows.

Lines 101-102: The author indicate that mutations in one MuvB core member *lin-37* result in ectopic expression of germline genes in the soma. However, loss of any of the five MuvB core proteins and also the DP homolog *dpl-1* result in ectopic expression of germline genes (Petrella 2011, Wu et al 2012). The way it is written it suggests that only loss of *lin-37* shows this result, which is inaccurate.

Thank you for bringing this to our attention. We have revised the sentence accordingly.

Figure 1C: It is really hard to see the double mutant with the same color as the single *daf-16* mutant- I suggest choosing a different contrasting color for the double mutant.

We have revised the colors to make them easier to distinguish.

Small suggested change Line 453-54: Original "though protein-phosphatase activity has not been demonstrated" change to "though protein-phosphatase activity is yet to be shown in *C. elegans*." Since the argument being made is that it does have this activity in *C. elegans* and the data presented suggests that- this is more compelling.

Thank you for this suggestion, which we have included.

March 26, 2025

RE: Life Science Alliance Manuscript #LSA-2024-03147-TR

Dr. L. Ryan Baugh
Duke University
Department of Biology
Box 90338
Durham, NC 27708

Dear Dr. Baugh,

Thank you for submitting your revised manuscript entitled "DAF-18/PTEN protects LIN-35/Rb from CLP-1/CAPN-mediated cleavage to promote starvation resistance". We would be happy to publish your paper in Life Science Alliance pending final revisions necessary to meet our formatting guidelines.

- please be sure that the authorship listing and order is correct
- please add the X and Bluesky handles of your host institute/organization as well as your own or/and one of the authors in our system
- Please incorporate any points from the Conclusion section into the Discussion; we only allow a Discussion section
- please add an Author Contributions section to your main manuscript text
- please add callouts for Figure S2A-B to your main manuscript text

A. FINAL FILES:

B. MANUSCRIPT ORGANIZATION AND FORMATTING:

Sincerely,

March 27, 2025

RE: Life Science Alliance Manuscript #LSA-2024-03147-TRR

Dr. L. Ryan Baugh
Duke University
Department of Biology
Box 90338
Durham, NC 27708

Dear Dr. Baugh,

Thank you for submitting your Research Article entitled "DAF-18/PTEN protects LIN-35/Rb from CLP-1/CAPN-mediated cleavage to promote starvation resistance". It is a pleasure to let you know that your manuscript is now accepted for publication in Life Science Alliance. Congratulations on this interesting work.

DISTRIBUTION OF MATERIALS:

Again, congratulations on a very nice paper. I hope you found the review process to be constructive and are pleased with how the manuscript was handled editorially. We look forward to future exciting submissions from your lab.

Sincerely,
